# Physics-Driven ML-Based Modelling for Correcting Inverse Estimation

**Ruiyuan Kang** *
Bayanat AI
Abu Dhabi, UAE
`ruiyuan.kang@bayanat.ai`

**Tingting Mu**
University of Manchester
Manchester, M13 9PL, United Kingdom
`tingting.mu@manchester.ac.uk`

**Panos Liatsis**
Khalifa Univeristy
Abu Dhabi, UAE
`panos.liatsis@ku.ac.ae`

**Dimitrios C. Kyritsis**
Khalifa Univeristy
Abu Dhabi, UAE
`dimitrios.kyritsis@ku.ac.ae`

## Abstract

When deploying machine learning estimators in science and engineering (SAE) domains, it is critical to avoid failed estimations that can have disastrous consequences, e.g., in aero engine design. This work focuses on detecting and correcting failed state estimations before adopting them in SAE inverse problems, by utilizing simulations and performance metrics guided by physical laws. We suggest to flag a machine learning estimation when its physical model error exceeds a feasible threshold, and propose a novel approach, GEESE, to correct it through optimization, aiming at delivering both low error and high efficiency. The key designs of GEESE include (1) a hybrid surrogate error model to provide fast error estimations to reduce simulation cost and to enable gradient based backpropagation of error feedback, and (2) two generative models to approximate the probability distributions of the candidate states for simulating the exploitation and exploration behaviours. All three models are constructed as neural networks. GEESE is tested on three real-world SAE inverse problems and compared to a number of state-of-the-art optimization/search approaches. Results show that it fails the least number of times in terms of finding a feasible state correction, and requires physical evaluations less frequently in general.

## 1 Introduction

Many estimation problems in science and engineering (SAE) are fundamentally inverse problem, where the goal is to estimate the state $\mathbf{x} \in \mathcal{X}$ of a system from its observation $\mathbf{y} \in \mathcal{Y}$. Examples include estimating the temperature state from the observed spectrum in combustion diagnostics [1], and discovering design parameters (state) of aero engine according to a group of performance parameters (observation) [2]. Traditional physics-driven inverse solvers are supported by rigorous physical laws, which vary depending on the application, e.g., the two-colour method for spectrum estimation [3], and cycle analysis for aero engine design [4]. Recent advances take advantage of machine learning (ML) techniques, constructing mapping functions $F$ to directly estimate the state from the observation, i.e., $\hat{\mathbf{x}} = F(\mathbf{y})$ [5, 6, 7]. Such ML solutions are more straightforward to develop, moreover, efficient and easy to use. However, ML-based state estimates can sometimes be erroneous, while SAE applications have very low error tolerance. One can imagine the disastrous

---

*The work is done during Ruiyuan's PhD and Postdoc at Khalifa Univeristy

consequences of providing unqualified aero engine design parameters. Therefore, it is critical to detect and correct failed ML estimations before adopting them.

This leads to a special SAE requirement of evaluating the estimation correctness in the deployment process of an ML estimator. Since the ground truth state is unknown at this stage, indirect evaluation has to be performed. Such evaluations can be based on physical forward models and performance metrics [8, 9]. A common practice is to combine multiple evaluations to obtain an accumulated physical error, enforcing quality control from different aspects.

When the physical error exceeds a feasibility threshold, one has to remediate the concerned ML estimation. One practice for finding a better estimation is to directly minimize the physical error in state space [10]. This requires solving a black-box optimization problem, for which it is challenging to find its global optimum, iterative approaches are used to find a near-optimal solution [11, 12]. In each iteration, a set of states are selected to collect their physical errors, then error feedback is used to generate better state(s) until a near-optimal state is found. Physical error collection involves time-consuming simulations[13, 14], e.g., a spectrum simulation which, despite taking just several minutes for each run [15], can become costly if queried many times. Consequently, the optimization process becomes time-consuming. Therefore, in addition to searching a satisfactory state with as small as possible physical error, it is also vital to decrease the query times to the physical evaluation.

Our work herein is focused on developing an efficient algorithm for remediating the concerned ML estimation in deployment. We propose a novel correction algorithm, **G**enerative **E**xploitation and **E**xploration guided by hybrid **S**urrogate **E**rror (GEESE), building upon black-box optimization. It aims at finding a qualified state within an error tolerance threshold after querying the physical evaluations as few times as possible. The key design elements of GEESE include: (1) A hybrid surrogate error model, which comprises an ensemble of multiple base neural networks, to provide fast estimation of the physical error and to enable informative gradient-based backpropagation of error feedback in model training. (2) A generative twin state selection approach, which consists of two generative neural networks for characterizing the distributions of candidate states, to effectively simulate the exploitation and exploration behaviours. We conduct thorough experiments to test the proposed algorithm and compare it with a series of state-of-the-art optimization/search techniques, based on three real-world inverse problems. Results show that, among the compared methods, GEESE is able to find a qualified state after failing the least number of times and needing to query the physical evaluations less times.

## 2 Related Work

**Optimization in SAE:** Development of SAE solutions often requires to formulate and solve optimization problems [16, 17, 18]. They are often black-box optimization due to the SAE nature. For instance, when the objective function is characterized through physical evaluations and solving partial differential equations (PDEs) [19], it is not given in a closed form. Typical black-box optimization techniques include Bayesian Optimization [20], Genetic Algorithm (GA) [21], and Particle Swarm Optimization (PSO) [22], etc. They often require a massive number of queries to the objective function in order to infer search directions for finding a near-optimal solution, which is time-consuming and expensive in SAE applications.

Instead, differentiable objective functions are constructed, and the problem is reduced to standard optimization, referred to as white-box optimization to be in contrast with black-box. A rich amount of well established solvers are developed for this, e.g., utilizing first-order and second-order gradient information [23]. Some recent developments use neural networks to optimize differentiable physical model evaluations, e.g., Optnet [24] and iterative neural networks [25]. However, physics-driven objective functions cannot always be formulated in a differential form, e.g., errors evaluated by the physical forward model in aero engine simulation, which is a mixture of database data, map information and PDEs [26]. A grey-box setting is thus more suitable in practice, where one does not overwrap the evaluations as a black box or oversimplify them as a white box, but a mixture of both.

**Surrogate Model in Black-box Optimization:** To reduce the cost of querying objective function values in black-box optimization, recent approaches construct surrogate models to obtain efficient and cheap estimation of the objective function. This practice has been used in SAE optimization, where the objective functions are mostly based on physical evaluations. The most popular technique for constructing surrogate models is machine learning (ML), including neural networks and Gaussian pro-

cess models [27, 28, 29]. The associated surrogate model is then incorporated within an optimization process, guided by, for instance, GA and Bayesian optimization, which generate states and interact with it [30, 29], or neural networks that work with differentiable surrogate models [31, 32, 12]. To avoid overfitting, recent effort has been invested to develop surrogate models consistent with some pre-collected data, aiming at obtaining more reliable near-optimal solutions [33, 34, 35, 36, 37]. Nevertheless, there is no guarantee that a surrogate model can well approximate a physical model consistently. Indeed, this is the motivation for the proposed method, where surrogate models are used to speed up the querying process, while the decision in regards to the suitability of the solution is based on the actual physical evaluation.

**Reinforcement Learning for Inverse Problems:** In addition to black-box optimization based approaches, Reinforcement Learning (RL) [38, 39] serves as an alternative framework for solving inverse problems [40, 41, 42]. In an RL-based solution framework, physical evaluations are wrapped as a black-box environment outputting scalar reward, and the actions are the states to estimate according to the observation. The behaviour of the environment is simulated by training a world/critic model [43, 44], which is equivalent to a surrogate model of the physical evaluations. Different from black-box optimization based approaches, RL does not intend to search a feasible state estimation for the given observation, but to learn an authoritative agent/policy model [45, 46] to provide state estimations, while the policy training is guided by optimizing an accumulated scalar reward or error [47, 48]. Because of the desire of training a powerful policy model and the statistical nature of the reward, RL often requires many physical evaluations to collect diverse samples and validate training performance [49, 50]. This can be time-consuming when there is limited computing resource.

## 3 Proposed Method

We firstly explain the notation convention: Ordinary letters, such as $x$ or $X$, represent scalars or functions with scalar output. Bold letters, such as $\mathbf{x}$ or $\mathbf{X}$, represent vectors or functions with vector output. The $i$-th element of $\mathbf{x}$ is denoted by $x_i$, while the first $k$ elements of $\mathbf{x}$ by $x_{1:k}$. We use $|\mathbf{x}|$, $\|\mathbf{x}\|_1$ and $\|\mathbf{x}\|_2$ to denote the dimension, $l_1$-norm and $l_2$-norm of the vector $\mathbf{x}$. An integer set is defined by $[n] = \{1, 2 \dots n\}$.

Without loss of generality, an estimated state $\hat{\mathbf{x}}$ is assessed by multiple physical models and/or metrics $\{P_i\}_{i=1}^h$, resulting to an $h$-dimensional error vector, denoted by

$$\mathbf{e}(\hat{\mathbf{x}}, \mathbf{y}) = [E_{P_1}(\hat{\mathbf{x}}, \mathbf{y}), E_{P_2}(\hat{\mathbf{x}}, \mathbf{y}), \dots, E_{P_h}(\hat{\mathbf{x}}, \mathbf{y})]. \tag{1}$$

Each concerned ML estimation obtained from an observation $\mathbf{y}$ is remediated independently, so $\mathbf{y}$ acts as a constant in the algorithm, which enables simplifying the error notation to $\mathbf{e}(\hat{\mathbf{x}})$ and $E_{P_i}(\hat{\mathbf{x}})$. A better state estimation is sought by minimizing the following accumulated physical error as

$$\min_{\hat{\mathbf{x}} \in \mathcal{X}} e(\hat{\mathbf{x}}) = \sum_{i=1}^h w_i E_{P_i}(\hat{\mathbf{x}}), \tag{2}$$

where the error weights are priorly identified by domain experts according to the targeted SAE application. For our problem of interest, the goal is to find a state correction that is within a desired error tolerance, e.g., $e(\hat{\mathbf{x}}) \le \epsilon$ where $\epsilon > 0$ is a feasibility threshold, determined by domain experts. Thus it is not necessary to find a global optimal solution, instead a feasible solution suffices. A typical iterative framework for black-box optimization can be used for this. For instance, at each iteration $t$, a set of selected states $\left\{\hat{\mathbf{x}}_i^{(t)}\right\}_{i=1}^{n_t}$ are queried to collect their physical errors resulting in a set of state-error pairs $\left\{\left(\hat{\mathbf{x}}_i^{(t)}, \mathbf{e}_i\right)\right\}_{i=1}^{n_t}$. A state analysis is then performed according to the error feedback. In the next iteration, a new set of states $\left\{\hat{\mathbf{x}}_i^{(t+1)}\right\}_{i=1}^{n_{t+1}}$ are selected to query. This process is repeated until the feasible state $\hat{\mathbf{x}}^*$ that satisfies $e(\hat{\mathbf{x}}^*) \le \epsilon$ is found. When designing such a framework, the objective is to find a feasible state $\hat{\mathbf{x}}^*$ by querying the physical errors as less times as possible because it is time-consuming to collect the errors.

We challenge the difficult setting of choosing only two states to query at each iteration. To ease the explanation, we first present a sketch of our proposed GEESE approach in Algorithm 1. It starts from an initial set of randomly selected and queried states $\left\{\left(\hat{\mathbf{x}}_i^{(0)}, \mathbf{e}_i\right)\right\}_{i=1}^N$. After this, at each

iteration, only two new states are selected to query based on a novel *twin state selection* approach that we propose, which selects a potentially near-optimal state for exploitation and a potentially informative state for exploration, resulting in $\left(\hat{\mathbf{x}}_{\text{IT}}^{(t)}, \mathbf{e}_{\text{IT}}^{(t)}\right)$ and $\left(\hat{\mathbf{x}}_{\text{R}}^{(t)}, \mathbf{e}_{\text{R}}^{(t)}\right)$. The selection requires to perform error analysis for a large set of candidate states, involving both the errors and their gradients. To ease and enable such computation, we develop a differentiable surrogate error model $\hat{e}\left(\mathbf{x}, \mathbf{w}\right)$ to rapidly approximate those error elements that are expensive to evaluate or in need of gradient calculation, and also provide informative gradient guidance with the assistance of error structure. The weights $\mathbf{w}$ of the surrogate model are trained using the queried state-error pairs, which start as $\left\{\left(\hat{\mathbf{x}}_i^{(0)}, \mathbf{e}_i\right)\right\}_{i=1}^{N}$ and then expand by including $\left(\hat{\mathbf{x}}_{\text{IT}}^{(t)}, \mathbf{e}_{\text{IT}}^{(t)}\right)$ and $\left(\hat{\mathbf{x}}_{\text{R}}^{(t)}, \mathbf{e}_{\text{R}}^{(t)}\right)$ at each iteration till the algorithm terminates by satisfying the feasibility condition $e(\hat{\mathbf{x}}^*) \leq \epsilon$. Below, we first explain the process of constructing the surrogate model for error approximation, followed by the twin state selection for characterizing the probability distributions of the candidate states and collecting errors, and finally, the implementation of the complete algorithm.

---

**Algorithm 1** Sketch of GEESE

---

1: Randomly select $n_0$ states and query their physical errors to obtain $D_0 = \left\{\hat{\mathbf{x}}_i^{(0)}, \mathbf{e}_i\right\}_{i=1}^{n_0}$
2: Train the surrogate error model $\hat{e}\left(\mathbf{x}, \mathbf{w}^{(0)}\right)$ with $D_0$
3: **for** $t \leq T$ **do**
4:    Select a query state $\hat{\mathbf{x}}_{\text{IT}}^{(t)}$ by exploitation and collect the state-error pair $\left(\hat{\mathbf{x}}_{\text{IT}}^{(t)}, \mathbf{e}_{\text{IT}}^{(t)}\right)$
5:    Update the feasible state $\hat{\mathbf{x}}^* = \hat{\mathbf{x}}_{\text{IT}}^{(t)}$
6:    Stop the algorithm if $e(\hat{\mathbf{x}}^*) \leq \epsilon$
7:    Select a query state $\hat{\mathbf{x}}_{\text{R}}^{(t)}$ by exploration and collect the state-error pair $\left(\hat{\mathbf{x}}_{\text{R}}^{(t)}, \mathbf{e}_{\text{R}}^{(t)}\right)$
8:    Expand the training data $D_t$ using the two pairs $\left(\hat{\mathbf{x}}_{\text{IT}}^{(t)}, \mathbf{e}_{\text{IT}}^{(t)}\right)$ and $\left(\hat{\mathbf{x}}_{\text{R}}^{(t)}, \mathbf{e}_{\text{R}}^{(t)}\right)$
9:    Keep training the surrogate error model using $D_t$, resulting in $\hat{e}\left(\mathbf{x}, \mathbf{w}^{(t)}\right)$
10: **end for**
11: output $\hat{\mathbf{x}}^*$

---

### 3.1 Hybrid Neural Surrogate Error Models

We start from an informal definition of implicit and explicit errors. Among the set of $h$ error elements in Eq. (1 ), those that are expensive to collect or to perform gradient calculation are referred to as *implicit errors*. These can include cases where the system is too complicated and needs much more time to calculate the gradient than that of network backpropagation; or where the system is indifferentiable, such as the physical model of spectroscopy [15] and aeroengine [26] containing database or map. In addition to implicit errors, the remaining are *explicit errors*. We order these error elements so that the first $k$ elements $\{E_{P_i}(\hat{\mathbf{x}})\}_{i=1}^{k}$ are implicit while the remaining $\{E_{P_i}(\hat{\mathbf{x}})\}_{i=k+1}^{n}$ are explicit. Our strategy is to develop a surrogate for each implicit error element, while directly calculate each explicit error.

Taking advantage of the robustness of ensemble learning [51, 52], we propose to estimate the implicit errors by an ensemble of multiple base neural networks. Each base neural network is fully connected with a mapping function $\phi(\mathbf{x}, \mathbf{w}) : \mathcal{R}^D \times \mathcal{R}^{|\mathbf{w}|} \to \mathcal{R}^k$, taking the $D$-dimensional state space $\mathcal{R}^D$ as its input space, while returning the approximation of the $k$ implicit errors by its $k$ output neurons. An example of such a state space is a space of $D = 2$ dimensions, where the two dimensions correspond to the temperature and concentration states from spectroscopy. Another example is a state space of $D = 11$ dimensions with each state corresponding to a design parameter for aeroengine design, for which we provide more details in Section 4 and Appendix A. The network weights are stored in the vector $\mathbf{w}$. We train $L$ individual base networks sharing the same architecture, while obtain the final prediction using an average combiner. As a result, given a state estimation $\hat{\mathbf{x}}$, the estimate of the implicit error vector is computed by

$$\hat{\mathbf{e}}_{\text{im}}\left(\hat{\mathbf{x}}, \{\mathbf{w}_i\}_{i=1}^{L}\right) = \frac{1}{L} \sum_{i=1}^{L} \phi\left(\hat{\mathbf{x}}, \mathbf{w}_i\right), \tag{3}$$

and thus, the accumulated physical error is approximated by

$$\hat{e}\left(\hat{\mathbf{x}}, \{\mathbf{w}_i\}_{i=1}^L\right) = \underbrace{\sum_{j=1}^k w_j \left(\frac{1}{L}\sum_{i=1}^L \phi_j\left(\hat{\mathbf{x}}, \mathbf{w}_i\right)\right)}_{\text{approximated implicit error}} + \underbrace{\sum_{j=k+1}^h w_j E_{P_j}(\hat{\mathbf{x}})}_{\text{true explicit error}}. \tag{4}$$

We refer to Eq. (4) as a hybrid surrogate error model including both approximated and true error evaluation.

The weights of the base neural networks $\{\mathbf{w}_i\}_{i=1}^L$ are trained using a set of collected state-error pairs, e.g., $D = \{(\hat{\mathbf{x}}_i, \mathbf{e}_i)\}_{i=1}^N$. In our implementation, bootstrapping sampling [53] is adopted to train each base neural network independently, by minimizing a distance loss between the estimated and collected implicit errors, as

$$\min_{\mathbf{w}_i} \mathbb{E}_{(\hat{\mathbf{x}},\mathbf{e})\sim D}\left[\text{dist}\left(\boldsymbol{\phi}\left(\hat{\mathbf{x}}, \mathbf{w}_i\right), \mathbf{e}_{1:k}\right)\right]. \tag{5}$$

A typical example of the distance function is $\text{dist}(\hat{\mathbf{e}}, \mathbf{e}) = \|\hat{\mathbf{e}} - \mathbf{e}\|_2^2$.

Here, we choose to estimate each element of the implicit error vector, rather than to estimate a scalar value of the weighted error sum, because the structural information of the error vector can directly contribute in training through its associated gradient information. When estimating the weighted sum directly, it is in a way to restrict the training loss to a form loosely like $(\hat{e}\left(\mathbf{w}\right) - \|\mathbf{e}\|_1)^2$, which negatively affects the information content of the gradient information. We have observed empirically that, the proposed individual error estimation leads to improvements in training the exploitation generator, compared to using the weighted error sum, see ablation study (1) in Table 2.

## 3.2 Twin State Selection

A selection strategy, i.e., twin state selection (TSS), for querying two individual states at each iteration is proposed, one for exploration and one for exploitation, respectively. The objective of TSS is to substantially reduce the cost associated with physical error collection. In turn, this translates to the formidable challenge of designing a selection process, which maximizes the informativeness of the associated physical error collection subject to minimizing query times. It is obviously impractical and inaccurate to adopt the naive approach of choosing directly one state by searching the whole space. Instead, we target at a two-folded task, researching (1) which candidate set of states to select from and (2) how to select.

By taking advantage of developments in generative AI, we construct generative neural networks to sample the candidate states. Specifically, we employ a latent variable $\mathbf{z} \in \mathcal{R}^d$, which follows a simple distribution, e.g., uniform distribution $\mathbf{z} \sim U\left([-a,a]^d\right)$, and a neural network $\mathbf{G}(\mathbf{z}, \boldsymbol{\theta})$ : $\mathcal{R}^d \times \mathcal{R}^{|\boldsymbol{\theta}|} \to \mathcal{R}^D$. The transformed distribution $p\left(\mathbf{G}(\mathbf{z}, \boldsymbol{\theta})\right)$ is then used to model the distribution of a candidate set. Thus, the task of candidate selection is transformed into determining the neural network weights $\boldsymbol{\theta}$ for the generator $\mathbf{G}$.

In general, exploitation attempts to select states close to the optimal one, whereas exploration attempts to select more informative states to enhance the error estimation. There are various ways to simulate the exploitation and exploration behaviours. For instance, in conventional black-box optimization, e.g., Bayesian optimization and GA, exploitation and exploration are integrated within a single state selection process [54], while in reinforcement learning, a balance trade-off approach is pursued [55, 39]. Our method treats them as two separate tasks with distinct strategies for constructing generators and selecting states.

**ExploITation:** To simulate the exploitation behaviour, the exploitation generator $\mathbf{G}_{\text{IT}}$ is trained at each iteration by minimizing the expectation of the physical error estimate, using the hybrid surrogate error model

$$\boldsymbol{\theta}_{\mathbf{G}_{\text{IT}}}^{(t)} = \arg\min_{\boldsymbol{\theta}\in\mathcal{R}^d} \mathbb{E}_{\mathbf{z}\sim U([-a,a]^d)}\left[\hat{e}\left(\mathbf{G}_{\text{IT}}(\mathbf{z}, \boldsymbol{\theta}), \left\{\mathbf{w}_i^{(t-1)}\right\}_{i=1}^L\right)\right], \tag{6}$$

where the base networks from the last iteration are used and we add the subscript $t-1$ to the weights of the error network for emphasizing. Finally, among the candidates generated by $\mathbf{G}_{\text{IT}}$ with its trained

weights $\boldsymbol{\theta}_{\mathbf{G}_{\mathrm{IT}}}^{(t)}$, we select the following state

$$\hat{\mathbf{x}}_{\mathrm{IT}}^{(t)} = \arg \min_{\hat{\mathbf{x}} \sim p\left(\hat{\mathbf{x}} | \boldsymbol{\theta}_{\mathbf{G}_{\mathrm{IT}}}^{(t)}\right)} \hat{e}\left(\hat{\mathbf{x}}, \left\{\mathbf{w}_i^{(t-1)}\right\}_{i=1}^{L}\right), \tag{7}$$

to query its physical error by Eq. (1), resulting in the state-error pair $\left(\hat{\mathbf{x}}_{\mathrm{IT}}^{(t)}, \mathbf{e}_{\mathrm{IT}}^{(t)}\right)$. If the queried error is less than the feasibility threshold, i.e., $\mathbf{e}_{\mathrm{IT}}^{(t)} \leq \epsilon$, this selected state is considered acceptable and the iteration is terminated. Otherwise, it is used to keep improving the training of the surrogate error model in the next iteration.

**ExploRation:** To simulate the exploration behaviour, a state that does not appear optimal but has the potential to complement the surrogate error model should be selected. We use an exploration generator $\mathbf{G}_{\mathrm{R}}$ to generate candidates. To encourage diversity so as to facilitate exploration, we assign the generator random weights sampled from a simple distribution, e.g.,

$$\boldsymbol{\theta}_{\mathbf{G}_{\mathrm{R}}}^{(t)} \sim N\left(0, \mathcal{I}^{|\theta_{\mathbf{G}_{\mathrm{R}}}|}\right). \tag{8}$$

We do not intend to train the exploration generator $\mathbf{G}_{\mathrm{R}}$, because any training loss that encourages exploration and diversity can overly drive the base networks to shift focus in the state space and cause instability in the integrated algorithm. Such an instability phenomenon, caused by training $\mathbf{G}_{\mathrm{R}}$, is demonstrated in the ablation study (2) in Table 2.

By adopting the idea of active exploration via disagreement [56, 57], we consider the state, for which the base networks are the least confident about to estimate the implicit errors, as more informative. Since we use an ensemble of base neural networks to estimate the error, the standard deviations of the base network predictions serve as natural confidence measures [56], which are stored in a $k$-dimensional vector:

$$\boldsymbol{\sigma}\left(\hat{\mathbf{x}}, \left\{\mathbf{w}_i^{(t-1)}\right\}_{i=1}^{L}\right) = \left[\sigma_1\left(\{\boldsymbol{\phi}_1\left(\hat{\mathbf{x}}, \mathbf{w}_i\right)\}_{i=1}^{L}\right), \ldots, \sigma_k\left(\{\boldsymbol{\phi}_k\left(\hat{\mathbf{x}}, \mathbf{w}_i\right)\}_{i=1}^{L}\right)\right]. \tag{9}$$

The state maximizing disagreement, i.e., an accumulated standard deviation, between the base networks, is selected given as

$$\hat{\mathbf{x}}_{\mathrm{R}}^{(t)} = \arg \max_{\hat{\mathbf{x}} \sim p\left(\hat{\mathbf{x}} | \boldsymbol{\theta}_{\mathbf{G}_{\mathrm{R}}}^{(t)}\right)} \boldsymbol{\sigma}\left(\hat{\mathbf{x}}, \left\{\mathbf{w}_i^{(t-1)}\right\}_{i=1}^{L}\right) \mathbf{w}_k^T, \tag{10}$$

where the row vector $\mathbf{w}_k = [w_1, w_2, \ldots, w_k]$ stores the implicit error weights. The state-error pair $\left(\hat{\mathbf{x}}_{\mathrm{R}}^{(t)}, \mathbf{e}_{\mathrm{R}}^{(t)}\right)$ is obtained after error collection.

**Surrogate Model Update:** To initialize the algorithm, we priorly collect a set of state-error pairs $D_0 = \{\mathbf{x}_i, \mathbf{e}_i\}_{i=1}^{N}$ for randomly selected states. Next, at each iteration $t$, two new states are selected and their physical errors are calculated, thus resulting in two new training examples to update the surrogate error model, and an expanded training set $D_t = D_{t-1} \cup \left(\hat{\mathbf{x}}_{\mathrm{IT}}^{(t)}, \mathbf{e}_{\mathrm{IT}}^{(t)}\right) \cup \left(\hat{\mathbf{x}}_{\mathrm{R}}^{(t)}, \mathbf{e}_{\mathrm{R}}^{(t)}\right)$. In our implementation, the base neural network weights $\mathbf{w}_i^{(t-1)}$ obtained from the previous iteration are further fine tuned using the two added examples $\left(\hat{\mathbf{x}}_{\mathrm{IT}}^{(t)}, \mathbf{e}_{\mathrm{IT}}^{(t)}\right)$ and $\left(\hat{\mathbf{x}}_{\mathrm{R}}^{(t)}, \mathbf{e}_{\mathrm{R}}^{(t)}\right)$, as well as the $N$ examples sampled from the previous training set $D_{t-1}$.

### 3.3 Remediation System and Implementation

Given an ML estimation $\hat{\mathbf{x}}$, the remediation system collects its physical error vector as in Eq. (1), then calculates the accumulated error from the objective function of Eq. (2) and compares it to the feasibility threshold $\epsilon > 0$. When the error exceeds the threshold, the GEESE algorithm is activated to search a feasible estimation $\hat{\mathbf{x}}^*$ such that $e\left(\hat{\mathbf{x}}^*\right) \leq \epsilon$ by querying the physical error as few times as possible. Algorithm 2 outlines the pseudocode of GEESE[2], while Fig.1 illustrates its system architecture. Our key implementation practice is summarized below.

---

[2]project repo: `https://github.com/RalphKang/GEESE`

---

**Algorithm 2** GEESE

---

**Require:** Physical error model (and or metrics) $\{P_i\}_{i=1}^h$ and error weights $\{w_i\}_{i=1}^h$, feasibility threshold $\epsilon > 0$, training frequency coefficient $\delta_G$, focus coefficient $c > 0$, maximum iteration numbers $T$ and $T_e$, early stopping threshold $\epsilon_e > 0$, initial training data size $N$
**Ensure:** An acceptable state $\hat{\mathbf{x}}^*$ with $e(\hat{\mathbf{x}}^*) \leq \epsilon$

1: **Initialize:** iteration index $t = 0$, initial base neural network weights $\left\{\mathbf{w}_i^{(0)}\right\}_{i=1}^L$, number of early stopped base neural networks $n_e = 0$, initial exploitation neural network weights $\boldsymbol{\theta}_{\mathbf{G}_{\text{IT}}}^{(0)}$
2: Sample $Z_{\text{IT}}$ for exploitation generator $\mathbf{G}_{\text{IT}}$
3: Randomly select $N$ states to query their physical errors to obtain $D_0 = \{\mathbf{x}_i, \mathbf{e}_i\}_{i=1}^N$
4: **for** $t \leq T$ **do**
5:     Set the added training dataset as $\Delta D_t = \emptyset$
6:     Update $\boldsymbol{\theta}_{\mathbf{G}_{\text{IT}}}^{(t)}$ by training with Eq. (6) approximated by $Z_{\text{IT}}$ for up to $T_G = \delta_G \lfloor \frac{2n_e}{L} + 1 \rfloor$ iterations
7:     Select exploitation query state $\hat{\mathbf{x}}_{\text{IT}}^{(t)}$ by Eq. (7) approximated by $X_{\text{IT}}^{(t)}$
8:     **if** $\hat{e}\left(\hat{\mathbf{x}}_{\text{IT}}^{(t)}, \left\{\boldsymbol{\theta}_i^{(t)}\right\}_{i=1}^L\right) \leq c\epsilon$ **then**
9:         Collect the new state-error pair $\Delta D_t = \Delta D_t \cup \left(\hat{\mathbf{x}}_{\text{IT}}^{(t)}, \mathbf{e}_{\text{IT}}^{(t)}\right)$, set $\hat{\mathbf{x}}^* = \hat{\mathbf{x}}_{\text{IT}}^{(t)}$
10:     **end if**
11:     **if** $e = \sum_{i=1}^h w_i E_{P_i}(\hat{\mathbf{x}}^*) \leq \epsilon$ **then**
12:         Stop the algorithm
13:     **end if**
14:     Sample $\boldsymbol{\theta}_{\mathbf{G}_{\text{R}}}^{(t)}$ by Eq. (8)
15:     Select exploration query state $\hat{\mathbf{x}}_{\text{R}}^{(t)}$ by Eq. (10) approximated by $X_{\text{R}}^{(t)}$
16:     Collect the new state-error pair $\Delta D_t = \Delta D_t \cup \left(\hat{\mathbf{x}}_{\text{R}}^{(t)}, \mathbf{e}_{\text{R}}^{(t)}\right)$, update the training data $D_t = D_{t-1} \cup \Delta D_t$
17:     Obtain $\tilde{D}_i$ by sampling randomly $N$ state-error pairs from $D_{t-1}$ for each base neural network. Prepare training datasets $\{D_i\}_{i=1}^L$ where $D_i = \Delta D_t \cup \tilde{D}_i$
18:     Update $\left\{\mathbf{w}_i^{(t)}\right\}_{i=1}^L$ by training each base neural network using $D_i$ by Eq. (5) for up to $T_e$ iterations, and count the number of early stopped base neural networks $n_e$
19: **end for**

---

**Empirical Estimation:** Eqs. (6), (7) and (10) require operations performed over probability distributions. In practice, we approximate these by Monte Carlo sampling. For Eq. (6), we minimize instead the average over the sampled latent variables $Z_{\text{IT}} = \{\mathbf{z}_i\}_{i=1}^{N_{\text{IT}}}$ with $\mathbf{z}_i \sim U\left([-a_{\text{IT}}, a_{\text{IT}}]^d\right)$, and this is fixed in all iterations. The search space of Eq. (7) is approximated by a state set computed from $Z_{\text{IT}}$ using the trained generator, i.e., $X_{\text{IT}}^{(t)} = \left\{\mathbf{G}_{\text{IT}}\left(\mathbf{z}_i, \boldsymbol{\theta}_{\mathbf{G}_{\text{IT}}}^{(t)}\right)\right\}_{i=1}^{N_{\text{IT}}}$. Similarly, the search space of Eq. (10) is approximated by a state sample $X_{\text{R}}^{(t)} = \left\{\mathbf{G}_{\text{R}}\left(\mathbf{z}_i, \boldsymbol{\theta}_{\mathbf{G}_{\text{R}}}^{(t)}\right)\right\}_{i=1}^{N_{\text{R}}}$ where $\mathbf{z}_i \sim U\left([-a_{\text{R}}, a_{\text{R}}]^d\right)$.

**Early Stopping:** When training the base neural networks for implicit error estimation, in addition to the maximum iteration number $T_e$, early stopping of the training is enforced when the training loss in Eq. (5) is smaller than a preidentified threshold $\epsilon_e$. As a result, a higher number $n_e$ of early stopped base neural networks indicates a potentially more accurate error estimation. This strengthens the confidence in training the generator $\mathbf{G}_{\text{IT}}$ by Eq. (6) that uses the trained base neural network from the previous iteration. In other words, when the base neural network are not sufficiently well trained, it is not recommended to put much effort in training the generator, which relies on the estimation quality. Therefore, we set the maximum iteration number $T_G$ for training $\mathbf{G}_{\text{IT}}$ in proportional to $n_e$, i.e., $T_G = \delta_G \lfloor \frac{2n_e}{L} + 1 \rfloor$, where $\delta_G$ is training frequency coefficient.

**Failed Exploitation Exclusion:** The state selection motivated by exploitation aims at choosing an $\hat{\mathbf{x}}_{\text{IT}}^{(t)}$ with comparatively low physical error. To encourage this, a focus coefficient $c$ is introduced,

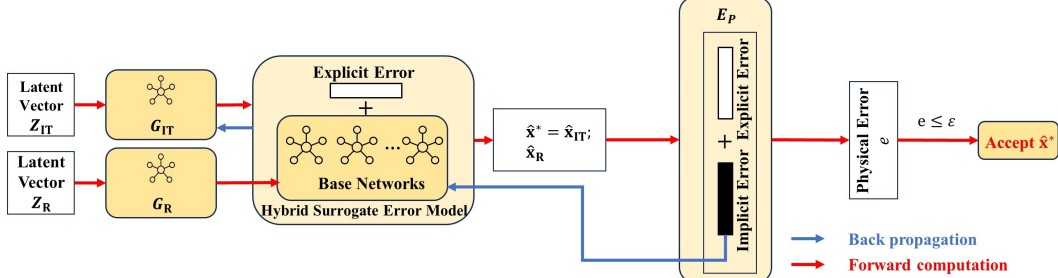

Figure 1: The workflow of GEESE: If the estimation from an ML estimator fails the physical evaluation $E_P$, GEESE is activated. The error estimated by hybrid surrogate error model is used to train the exploitation generator $\mathbf{G}_{\text{IT}}$. Two candidate state sets are generated by $\mathbf{G}_{\text{IT}}$ and exploration generator $\mathbf{G}_{\text{R}}$, and finally, two states $\hat{\mathbf{x}}^* = \hat{\mathbf{x}}_{\text{IT}}$ and $\hat{\mathbf{x}}_{\text{R}}$ are selected by the surrogate error model and fed into $E_P$ for evaluation and data collection. The process is terminated till $e(\hat{\mathbf{x}}^*) \leq \epsilon$.

which, together with the feasibility error threshold $\epsilon > 0$, is used to exclude a potentially failed state with a high estimated error, i.e., $\hat{e}\left(\hat{\mathbf{x}}, \{\mathbf{w}_i\}_{i=1}^{L}\right) > c\epsilon$, to avoid an unnecessary query.

## 4 Experiments and Results

We test the proposed approach GEESE on three real-world engineering inverse problems, including aero engine design [42], electro-mechanical actuator design [58] and pulse-width modulation of 13-level inverters [59]. The first problem is to find eleven design parameters (state) of an aero engine to satisfy the thrust and fuel consumption requirement (observation). The second problem is to find 20 design parameters (state) of an electro-mechanical actuator to satisfy requirements for overall cost and safety factor (obversation). And the third problem is to find a group of 30 control parameters (state) of a 13-level inverter to satisfy the requirements for distortion factor and nonlinear factor (observation). Details of these problems along with their physical models and metrics for evaluation are explained in supplementary material (Section A). We compare it with a set of classical and state-of-the-art black-box optimization techniques, including Bayesian Optimization with Gaussian Process (BOGP), GA [21], PSO [22], CMAES [60], ISRES [61], NSGA2 [62], and UNSGA3 [63], as well as the recently proposed work SVPEN [42], which employs RL in solving SAE inverse problems. These techniques are chosen because they are effective at seeking solutions with the assist of actual physical evaluations.

In practice, different types of simulators exist for the same problem. For instance, for problem 1, a simulator mentioned in [64] with high fidelity takes 10.3 seconds $\times$ 309 iterations = 53 minutes to obtain a converged simulation, while another simulator in [65] with a much lower fidelity can realize real-time simulation. Since we aim at a research scenario that attempts to search a feasible state without querying too much data and without setting a high standard on data quality, we choose to use faster simulators with lower but reasonable fidelity in our experiments. For the three studied problems, each simulator takes no more than five seconds to run. Since the computing time varies when changing simulators, we report the performance by query times instead of actual computing time. Accordingly, we adopt two metrics to compare performance. First, we set a maximum budget of $T = 1,000$ query times for all studied problems and compared methods, and test each method on each problem individually with 100 experimental cases, each case corresponds to a concerned ML state estimation. The setup of the experimental cases is described in Appendix A of supplementary material. we measure the number of experiments out of 100 where a method fails to correct the concerned estimation when reaching the maximum query budget, and refer to it as the failure times $N_{\text{failure}}$. Also, the average number of queries that a method requires before finding a feasible state in an experiment, is reported over 100 experiments, and referred to as average query times $N_{\text{query}}$. A more competitive algorithm expects smaller $N_{\text{failure}}$ and $N_{\text{query}}$.

We report the adopted hyper-parameter and model setting for GEESE: The common hyperparameter settings shared between all three studied problems include $T_e = 40$, $\epsilon_e = 1e^{-4}$ and $N = 64$, and the learning rates of $1e^{-2}$ and $1e^{-4}$, for training the exploitation generator and base neural networks, respectively. Different focus coefficients of $c = 1.5, 2$ and $5$ (set in an increasing fashion) are used for problems 1, 2 and 3, respectively, due to an increased problem complexity in relation to their

Table 1: Performance comparison of the compared methods, where the best is shown in **bold**, while the second best is underlined

| Algorithm | Problem 1 State Dimension:11 | | Problem 2 State Dimension:20 | | Problem 3 State Dimension:30 | |
|---|---|---|---|---|---|---|
| | Failure times | Query times | Failure times | Query times | Failure times | Query times |
| BOGP | 0 | 3.29 ±1.51 | 97 | 973.76 ±144.28 | 4 | 112.66 ±229.98 |
| GA | 0 | 64.00 ±0.00 | 0 | 130.56 ±63.31 | 13 | 231.76 ±339.71 |
| PSO | 0 | 64.00 ±0.00 | 0 | 64.00 ±0.00 | 12 | 244.16±343.71 |
| CMAES | 0 | 55.67 ±3.28 | 0 | 119.44 ±41.80 | 12 | 227.42 ±312.17 |
| ISRES | 0 | 65.00±0.00 | 0 | 177.64 ±80.51 | 16 | 250.05 ±350.16 |
| NSGA2 | 0 | 64.00 ±0.00 | 0 | 139.52 ±68.56 | 13 | 232.40 335.94 |
| UNSGA3 | 0 | 64.00 ±0.00 | 0 | 140.80 ±79.94 | 12 | 227.52 ±330.07 |
| SVPEN | 100 | 1000.00±0.00 | 100 | 1000.00±0.00 | 100 | 1000.00±0.00 |
| GEESE (Ours) | 0 | **3.18 ±1.98** | 0 | **51.65 ±33.01** | **0** | **43.56 ±65.28** |

increasing dimensions of the state space. Similarly, an increasing training frequency coefficient $\delta_G = 1, 1$ and 7 is used for problems 1, 2 and 3, respectively, because the problem requires more training iterations as it involves more complex patterns from higher-dimensional state space. The ensemble surrogate model for estimating the implicit errors is constructed as an average of 4 multi-layer perceptrons (MLPs) each with three hidden layers consisting of 1024, 2028 and 1024 hidden neurons. The exploration generator $\mathbf{G}_R$ is constructed as a single layer perceptron (SLP) and its one-dimensional input is sampled from $U([-5, 5])$. For problems 1 and 2 that are relatively less complex from an engineering point of view, we directly set the latent space $\mathcal{Z}$ as the state space $\mathcal{X}$ without using any neural network to transform in between. Then, we directly sample initial state set $X_{\mathrm{IT}}^{(0)}$. The exploitation state is directly optimized iteratively, e.g., by a gradient descent approach based on Eq.(7) to obtain state set $X_{\mathrm{IT}}^{(t)}$, as shown in Eq. (11). The one with the smallest objective function value is selected as the exploitation state, i.e.,

$$\hat{\mathbf{x}}_{\mathrm{IT}}^{(t)} = \arg \min_{\hat{\mathbf{x}} \in \mathcal{X}} \hat{e}\left(\hat{\mathbf{x}}, \left\{\mathbf{w}_i^{(t-1)}\right\}_{i=1}^{L}\right). \tag{11}$$

Problem 3 involves a special state pattern, requiring an increasing state value over the dimension, i.e., $\mathbf{x}_i - \mathbf{x}_{i+1} < 0$. To enable the latent variables to capture this, we construct the exploitation generator $\mathbf{G}_{\mathrm{IT}}$ as an MLP with three hidden layers consisting of 256, 512 and 256 hidden neurons. Also, to avoid generation collapse [66] in problem 3, a regularization term has been added to the training loss in Eq. (6), resulting in the following revised training to encourage state diversity, as

$$\boldsymbol{\theta}_{\mathbf{G}_{\mathrm{IT}}}^{(t)} = \arg \min_{\boldsymbol{\theta} \in \mathcal{R}^{30}} \mathbb{E}_{\mathbf{z} \sim U\left([-5,5]^{30}\right)} \left[\hat{e}\left(\mathbf{G}_{\mathrm{IT}}(\mathbf{z}, \boldsymbol{\theta}), \left\{\mathbf{w}_i^{(t-1)}\right\}_{i=1}^{L}\right) + \max\left(0.0288 - \sigma_1(\mathbf{z}, \boldsymbol{\theta}), 0\right)\right], \tag{12}$$

where $\sigma_1(\mathbf{z}, \boldsymbol{\theta})$ denotes the standard deviation of the first state element generated by $\mathbf{G}_{\mathrm{IT}}$. We encourage it to shift away from the collapsed point but not overly spread, by bounding $\sigma_1$ with a portion of the standard deviation of a uniform distribution, e.g., 0.288, and the portion $\frac{0.288}{10} = 0.0288$ is observed empirically effective. The spread control is only needed for the first state as the remaining states follow by $\mathbf{x}_i - \mathbf{x}_{i+1} < 0$. Configurations of the competing methods, together with extra information on GEESE, are provided in Appendix B of supplementary material.

### 4.1 Results and Comparative Analysis

Table 1 summarizes the results of the compared methods for the three problems, obtained with a feasibility threshold of $\epsilon = 0.075$, which reflects high challenge with low error tolerance. It can be observed that GEESE has the least failure times $N_{\mathrm{failure}}$ on all three problems. In problem 3, especially, GEESE succeeds with no failure while most other methods have more than 10 failures. This is a highly desired characteristic for a remediation system with low error tolerance. In addition, GEESE also has the least query times $N_{\mathrm{query}}$ in all three problems, indicating the best efficiency. We report additional results in Appendix C of supplementary material by varying the feasibility threshold $\epsilon$ and the initial sample size $N$, where GEESE also achieves satisfactory performance in general, while outperforming other methods in handling higher-dimensional problems with lower error tolerance. SVPEN [42] cannot return a feasible correction in 1000 queries in all experiments, as its core supporting RL requires a lot more queries than other optimization based techniques.

## 4.2 Ablation Studies and Sensitivity Analysis

To examine the effectiveness of the key design elements of GEESEE, we perform a set of ablation studies and report the results in Table 2 using problem 1 with a small feasibility threshold $\epsilon = 0.05$ indicating low error tolerance. The studies include the following altered designs: (1) Estimate directly the implicit error sum using an MLP with the same hidden layers but one single output neuron. (2) Train the exploration generator $G_R$ by using an approach suggested by [57]. (3) Remove the early stopping design. (4) Remove the focus coefficient.

Results show that estimating the implicit error sum worsens the performance. As explained earlier in Section 3.1, this is because the structural information in gradient is lost in error sum estimation, causing ambiguous update when training $G_{IT}$, and consequently requires GEESE to make more error queries. Also training $G_R$ worsens the performance as compared to just assigning random network weights to $G_R$ without training. As previously explained in Section 3.2, this is because training $G_R$ can frequently shift the focus of the surrogate error model and, thus, impact on the stability of the optimization process. Both early stopping and focus coefficient play an important role in GEESE, where the former prevents GEESE from overfitting and the latter helps avoid unnecessary queries. Additional results on hyperparameter sensitivity analysis for GEESE are provided in Appendix D of supplementary material. The results show that GEESE is not very sensitive to hyperparameter changes and allows a wide range of values with satisfactory performance, which makes GEESE easy to be tuned and used in practice.

Table 2: Results of ablation studies reported on problem 1, where a better performance is highlighted in **bold**.

| (1): Individual vs Sum Error Estimation | | |
| --- | --- | --- |
| Surrogate Error Model | Query times | Standard deviation |
| Estimate error elements | **20.20** | **16.37** |
| Estimate error sum | 23.26 | 21.18 |

| (2): Effect of Exploration Training | | |
| --- | --- | --- |
| Exploration style | Query times | Standard deviation |
| w/o training | **32.64** | **22.82** |
| with training | 41.32 | 97.15 |

| (3): Effect of Early stopping | | |
| --- | --- | --- |
| Schedule | Query times | Standard deviation |
| with earlystop | **20.20** | **16.37** |
| w/o earlystop | 32.80 | 17.84 |

| (4): Effect of Focus Coefficient | | |
| --- | --- | --- |
| Schedule | Query times | Standard deviation |
| with focus coefficient | **20.20** | **16.37** |
| w/o focus coefficient | 27.19 | 19.36 |

## 5 Discussion and Conclusion

We have proposed a novel physics-driven optimization algorithm GEESE to correct ML estimation failures in SAE inverse problems. To query less frequently expensive physical evaluations, GEESE uses a cheaper hybrid surrogate error model, mixing an ensemble of base neural networks for implicit error approximation and analytical expressions of exact explicit errors. To effectively model the probability distribution of candidate states, two generative neural networks are constructed to simulate the exploration and exploitation behaviours. In each iteration, the exploitation generator is trained to find the most promising state with the smallest error, while the exploration generator is randomly sampled to find the most informative state to improve the surrogate error model. These two types of selection are separately guided by the approximated error by the ensemble and the disagreement between its base neural networks. The element-wise error approximation promotes a more effective interaction between the surrogate error model and the two generators. Being tested on three real-world engineering inverse problems, GEESE outperforms all the compared methods, finding a feasible state with the least query number with no failure under the low error tolerance setup.

However, there are still challenges to address in the future, particularly for very high-dimensional inverse problems. Such problems are in need of larger and more complex model architecture to accommodate their more complex underlying patterns, and thus impose challenge on training time and data requirement. Computation expense should not only consider the query cost of physical evaluations but also the learning cost of such models. Flexible neural network architectures that allow for embedding domain specific or induced knowledge in addition to simulation data and their training, as well as their interaction with the main solution model, e.g., an ML estimator for inverse problems, are interesting directions to pursue.

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

# A Studied Inverse Problems

## A.1 Problem Description

**Problem 1, Turbofan Design:** Turbofan is one of the most complex gas aero engine systems, and it is the dominant propulsion system favoured by commercial airliners [26]. The inverse problem for turbofan design is to find a group of design parameters modelled as the state to achieve the desired performance modelled as observation. The observation includes two performance parameters, including the thrust $y_t$ and the thrust specific fuel consumption $y_f$. The state includes 11 design parameters that control the performance of the engine, including the bypass ratio $r_{bp}$, the fan pressure ratio $\pi_{fan}$, the fan efficiency $\eta_{fan}$, the low-pressure compressor pressure ratio $\pi_{LC}$, the low-pressure compressor efficiency $\eta_{LC}$, the high-pressure compressor pressure ratio $\pi_{HC}$, the high-pressure compressor efficiency $\eta_{HC}$, the combustor efficiency $\eta_{LC}$, the combustion temperature in the burner $T_B$, the efficiency of high-pressure turbine $\eta_{HT}$, and the efficiency of low-pressure turbine $\eta_{LT}$. Following the same setting as in [42], the goal is to estimate the design parameters that can achieve the performance of a CFM-56 turbofan engine, for which the thrust should be 121 KN and the thrust specific fuel consumption should be 10.63 g/(kN.s) [67]. This corresponds to the observation vector $\mathbf{y} = [y_t, y_f] = [121, 10.63]$. The 100 experiment cases tested on this problem differ from the state to correct, which is randomly sampled from the feasible region of the design parameter space provided by [42]. Table 3 reports the allowed range of each design parameter, which all together define the feasible region.

Table 3: Feasible region of the design parameter space for problem 1.

| Range | $r_{bp}$ | $\pi_{fan}$ | $\pi_{LC}$ | $\pi_{HC}$ | $T_B$ | $\eta_{fan}$ | $\eta_{HC}$ | $\eta_{LC}$ | $\eta_B$ | $\eta_{HT}$ | $\eta_{LT}$ |
|---|---|---|---|---|---|---|---|---|---|---|---|
| Min | 5 | 1.3 | 1.2 | 8 | 1300 K | 0.85 | 0.82 | 0.84 | 0.95 | 0.86 | 0.87 |
| Max | 6 | 2.5 | 2 | 15 | 1800 K | 0.95 | 0.92 | 0.94 | 0.995 | 0.96 | 0.97 |

**Problem 2, Electro-mechanical Actuator Design:** An electro-mechanical actuator is a device that converts electrical energy into mechanical energy [68], by using a combination of an electric motor and mechanical components to convert an electrical signal into a mechanical movement. It is commonly used in industrial automation[68], medical devices[69], and aircraft control systems [70], etc. We consider the design of an electro-mechanical actuator with a three-stage spur gears. Its corresponding inverse problem is to find 20 design parameters modelled as the state, according to the requirements for the overall cost $y_c$ and safety factor $y_s$ modelled as the observation. The 100 experiment cases tested on this problem differ from the observation $\mathbf{y} = [y_c, y_s]$. We have randomly selected 100 combinations of the safety factor and overall cost from the known Pareto front [58], which is shown in Fig. 2a. For each observation, the state to correct is obtained by using an untrained ML model to provide a naturally failed design.

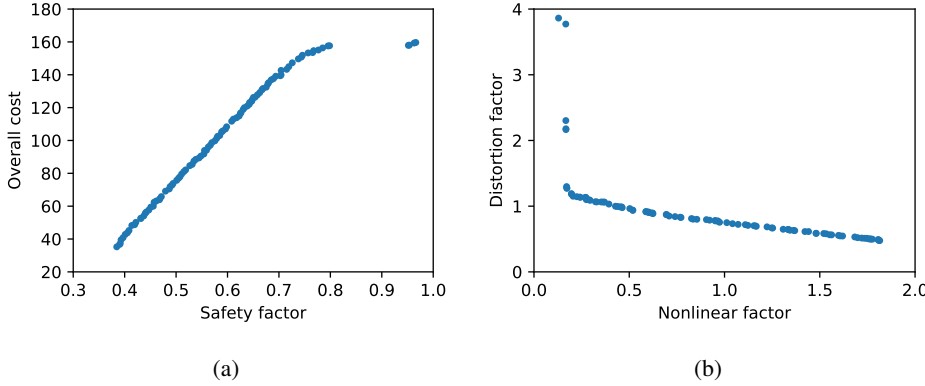

(a)                                                (b)

Figure 2: Illustration of the 100 used test cases in the 2-dimensional observation space for problem 2 (subfigure a) and 3 (subfigure b).

**Problem 3, Pulse-width Modulation of 13-level Inverters:** Pulse-width modulation (PWM) of n-level inverters is a technique for controlling the output voltage of an inverter that converts DC power into AC power [71]. It modulates the duty cycle of the output waveform of the inverter,

thereby affecting the effective value of the voltage applied to a load. Particularly, an PWM of 13-level inverter adjusts its output waveform using 13 power switching devices to achieve higher precision, which is widely used in renewable power generation systems [72], electric vehicles [73], and industrial automation equipment [74]. It results in a typical inverse problem of finding the suitable control parameters including 30 switch angles modelled as the state, according to the requirements of the distortion factor $y_d$ (which measures the harmonic distortion of the output waveform) and the nonlinear factor $y_n$ (which avoids the malfunctioning of the inverter and the connected load) modelled as the observation. As in problem 2, the 100 experiment cases tested on this problem also differ from the observation, i.e. $\mathbf{y} = [y_d, y_n]$. They correspond to 100 randomly selected combinations of the distortion and nonlinear factors from the known Pareto front in [59], which are shown in Fig. 2b. For each observation, the state to correct is obtained by using an untrained ML model to provide a naturally failed estimation.

## A.2 Physical Evaluation

We describe in this section how the physical evaluations are conducted, more specifically, how the physical errors are assessed. Overall, it includes the *observation reconstruction error*, which is based on the difference between the given observation and the reconstructed observation from the estimated state. For different problems, different physical models are used to simulate the reconstruction. It also includes the *feasible domain error*, which examines whether the estimated state is within a feasible region of the state space, and this region is often known for a given engineering problem. Apart from these, there are also other problem-specific errors.

### A.2.1 Problem 1

**Observation Reconstruction Error:** The gas turbine forward model [42] is used to simulate the performance of the turbofan engine. It is constructed through the aerodynamic and thermodynamic modelling of the components in a turbofan engine, where the modelled components include the inlet, fan, low-pressure and high-pressure compressors, combustor, high-pressure and low-pressure turbines, core and fan nozzles, as well as through considering the energy losses. This model can transform the input of state into physically reasonable output of observation, which is the thrust $y_t$ and fuel flow $y_f$ of the engine. Let $F(\mathbf{x})$ denote a forward model. In problem 1, the performance requirement is specifically $\mathbf{y} = [y_t, y_f] = [121, 10.63]$, thus, for a estimated state $\hat{\mathbf{x}}$, the reconstruction error is

$$e_r(\hat{\mathbf{x}}) = \sum_{i=1}^{2} \frac{||F_i(\hat{\mathbf{x}}) - \mathbf{y}_i||_1}{2\mathbf{y}_i}, \tag{13}$$

where, when $i$ respectively equals to 1 or 2, $F_1(\hat{\mathbf{x}})$ and $F_2(\hat{\mathbf{x}})$ are the estimated thrust and fuel consumption in the engine case, respectively. Because the magnitude of the thrust and fuel consumption are different, we use the relative error to measure the reconstruction error of the two observation elements.

**Feasible Domain Error:** In aero engine design, the design parameters cannot exceed their feasible region and such a region has already been identified by existing work [42] as in Table 3. For the $i$-th dimension of an estimated state $\hat{x}_i$ (an estimated design parameter), and given its maximum and minimum allowed values $x_{\max}$ and $x_{\min}$, we define its feasible domain error by

$$e_i^{(f)} = \max\left(\frac{\hat{\mathbf{x}}_i - x_{\min}}{x_{\max} - x_{\min}} - 1, 0\right) + \max\left(-\frac{\hat{\mathbf{x}}_i - x_{\min}}{x_{\max} - x_{\min}}, 0\right). \tag{14}$$

After normalization, all the feasible values are within the range of $[0, 1]$, while the non-feasible ones outside. The above error simply examines how much the normalized state value exceeds 1 or below 0. We compute an accumulated feasible error for all the 11 design parameters, given by $e_f(\hat{\mathbf{x}}) = \frac{1}{11} \sum_{i=1}^{11} e_i^{(f)}$.

**Design Balance Error:** Another desired property by aero engine design is a low disparity among the values of the design parameters after normalizing them by their feasible ranges, which indicates a more balanced design, offering better cooperation between the design components and resulting in lower cost [42, 26]. Standard deviation is a suitable measure to assess this balance property, resulting in another physical error

$$e_\sigma(\hat{\mathbf{x}}) = \sigma\left(\left\{\frac{\hat{\mathbf{x}}_i - x_{\min}}{x_{\max} - x_{\min}}\right\}_{i=1}^{11}\right). \tag{15}$$

where $\sigma(\cdot)$ denotes the standard deviation of the elements from its input set.

**Accumulated Physical Error:** The above three types of errors are combined to form the following accumulated physical error:

$$\hat{e}(\hat{\mathbf{x}}) = e_r(\hat{\mathbf{x}}) + 0.1e_f(\hat{\mathbf{x}}) + 0.1e_\sigma(\hat{\mathbf{x}}). \tag{16}$$

The weights are given as 1, 0.1 and 0.1, respectively. This is because the reconstruction error determines whether the estimated state is feasible, while the other two errors are used to further improve the quality of the estimated state from the perspective of the design preference. Here $e_r(\hat{\mathbf{x}})$ is obtained using a forward simulation process thus is an implicit error, while $\mathbf{e}_f$ and $e_\sigma$ have analytical expressions and simple gradient forms, and thus are explicit errors.

### A.2.2 Problem 2

**Observation Reconstruction Error**: The used forward model for electro-mechanical actuator design is a performance simulation model, considering a stepper motor, three stages of spur gears and a housing to hold the components (i.e., stepper motor, and three stages of spur gears) [58]. It consists of a physical model that predicts its output speed and torque and component-specific constraints, a cost model and a geometric model that creates 3-D meshes for the components and the assembled system. The integrated model predicts the observation $\mathbf{y} = \{y_c, y_s\}$, and is named as the "CS1" model in [58]. After reconstructing by CS1 the safety factor $y_s$ and total cost $y_c$ from the estimated design parameters $\hat{\mathbf{x}}$, the reconstruction error is computed using Eq. 13.

**Feasible Domain Error**: The same feasible domain error $\mathbf{e}_f$ as in Eq. (14) is used for each design parameter of problem 2. The only difference is that the allowed parameter ranges for defining the feasible region have changed. We use the region identified by [58]. There are 20 design parameters, thus $e_f$ is an average of 20 individual errors.

**Inequality Constraint Error**: We adopt another seven inequality constraints provided by the forward model [58] to examine how reasonable the estimated design parameters are. These constraints do not have analytical forms, and we express them as $c_i(\hat{\mathbf{x}}) \le 0$ for $i = 1, 2, , \dots, 7$. Based on these, we define the following inequality constraint error

$$e_c(\hat{\mathbf{x}}) = \frac{1}{7}\sum_{i=1}^{7} \max(c_i(\hat{\mathbf{x}}), 0). \tag{17}$$

**Accumulated Physical Error:** We then combine the above three types of errors, given as

$$\hat{e}(\hat{\mathbf{x}}) = e_r(\hat{\mathbf{x}}) + 0.1e_f(\hat{\mathbf{x}}) + e_c(\hat{\mathbf{x}}), \tag{18}$$

where both $e_r(\hat{\mathbf{x}})$ and $e_c(\hat{\mathbf{x}})$ are implicit errors computed using a black-box simulation model, while $\mathbf{e}_f$ is an explicit error. In this case, we increase the weight for inequality constraint error to be the same as the reconstruction error, this is because we regard the implicit errors irrespective of their types as the same. Of course, one can also use different weights for different types of errors according to their expertise.

### A.2.3 Problem 3

We use the forward model from [59] to reconstruct the observation for the 13-level inverter. It takes the control parameters as the input and returns the distortion factor $y_d$ and the nonlinear factor $y_n$. Based the reconstructed $y_d$ and $y_n$, the observation reconstruction error $e_r$ is computed by Eq. (13) in the same way as in problems 1 and 2. Similarly, the same feasible boundary error $e_f$ as in Eq. (14) is computed, but the feasible region is different where the range of $[0, \frac{\pi}{2}]$ is applied for all the 30 control parameters, which is defined in [59]. A similar inequality constraint error as in Eq. (17) is used, which contains 29 inequality constraints in the form of

$$c_i(\hat{\mathbf{x}}) = \hat{\mathbf{x}}_i - \hat{\mathbf{x}}_{i+1} < 0, \text{for } i = 1, 2, \dots 29. \tag{19}$$

Finally, the accumulated physical error is given by

$$\hat{e}(\hat{\mathbf{x}}) = e_r(\hat{\mathbf{x}}) + 0.1e_f(\hat{\mathbf{x}}) + 10e_c(\hat{\mathbf{x}}), \tag{20}$$

where a large weight is used for $e_c(\hat{\mathbf{x}})$ because the inequality constraints that it involves are very critical for the design. Among the three types of errors, $e_r(\hat{\mathbf{x}})$ is an implicit error, while $e_f(\hat{\mathbf{x}})$ and $e_c(\hat{\mathbf{x}})$ are explicit errors.

# B  Extra Implementation Information

In this section, we introduce extra implementation information for GEESE and the compared methods, in addition to what has been mentioned in the main text. In GEESE implementation, the latent vector $\mathbf{z}$ has the same dimension as the state $\mathbf{x}$ in problems 1 and 2, because the optimization is done directly on the latent vectors. In problem 3, the dimension of $\mathbf{z}$ is set be 1, and transformed into a 30-dimensional vector $\mathbf{x}$ by the state generator. The number of the latent vector $\mathbf{z}$ used for sampling distribution of generators is set increasingly as $d = 64, 128, 256$ for problems 1, 2, and 3, due to the increasing dimension of the state space of the three problems. Although, the budget query number equals to 1000, because GEESE may query two times per iteration, thus, the maximum iteration number is smaller than 1000, which is determined when the budget is used up.

For BOGP, its Bayesian optimization is implemented using the package [11]. The prior is set to be a Gaussian process, and its kernel is set as Matern 5/2. The acquisition function is set to be the upper confidence bound (UCB). The parameter kappa, which indicates how closed the next parameters are sampled, is tuned and the optimal value is 2.5. The other hyperparameters are kept as default. Since Bayesian optimization only queries one state-error pair in each iteration, its maximum iteration number is equal to the maximum number of queries, i.e., 1000.

The other methods of GA, PSO, CMAES, ISRES, NSGA2, and UNSGA3 are implemented using the package pymoo [75]. For ISRES, we optimize the offspring size, and finally apply a $1/7$ success rule to generate seven times more candidates than that in the current population in order to perform a sufficient search. The other parameters are kept as default. Since these algorithms need to query the whole population in each iteration, their maximum iteration number is thus much smaller than the query budget 1000. In the experiments, these algorithms are terminated when the maximum query number 1000 is reached.

To implement SVPEN [42], we use the default setting for problem 1. As for problems 2 and 3, to construct the state estimator and the error estimator for SVPEN, the same structures of the base neural networks and the exploitation generator as used by GEESE are adopted, respectively. Also the same learning rate as used by GEESE is used for SVPEN, while the other settings are kept as default for problems 2 and 3. In each iteration, SVPEN queries three times the physical errors for simulating the exploitation, as well as the regional and global exploration. Thus, the maximum iteration number of SVPEN is set as 333 to match the query budget 1000.

All the methods are activated or initialized using the same set of $N$ state-error pairs randomly sampled from a predefined feasible region in the state space. For GEESE and SVPEN, these samples are used to train their surrogate error models, i.e., the base neural networks in GEESE and the error estimator in SVPEN, thus their batch size for training is also set as $N$. For Bayesian optimization, these samples are used to construct the Gaussian process prior. For GA, PSO, ISRES, NSGA2, and UNSGA3, these samples are used as the initial population to start the search. The only special case is CMAES, as it does not need a set of samples to start the algorithm, but one sample. So we randomly select one state-error pair from the $N$ pairs to activate its search.

For problem 3, we post-process the output of all the compared methods, in order to accommodate the element-wise inequality constraints in Eq. (19), by

$$\hat{\mathbf{x}}_i^{(p)} = \hat{\mathbf{x}}_1^{(p)} + \frac{1}{1 + e^{-\sum_{j=1}^i \hat{\mathbf{x}}_j^{(p)}}} \left(1 - \hat{\mathbf{x}}_1^{(p)}\right). \tag{21}$$

As a result, the magnitude of the element in $\hat{\mathbf{x}}^{(p)}$ is monotonically increasing, and the inequality constraints are naturally satisfied. But this can complicate the state search, as the elements are no longer independent. A balance between correlating the state elements and minimizing the accumulated physical error is needed. But in general, we have observed empirically that the above post-processing can accelerate the convergence for all the compared methods. One way to explain the effectiveness of this post-processing is that it forces the inequality constraints to hold, and this is hard for the optimization algorithms to achieve on their own.

# C  Extra Results

**Varying Feasibility Threshold:** In addition to the feasibility threshold of $\epsilon = 0.075$ as studied in the main text, we test two other threshold values, including $\epsilon = 0.05$ representing a more challenging

Table 4: Performance comparison for two different values of feasibility threshold $\epsilon$, where the best is shown in **bold** while the second best is underlined for query times.

| Threshold | Algorithm | Problem 1 | | Problem 2 | | Problem 3 | |
|---|---|---|---|---|---|---|---|
| | | State Dimension:11 | | State Dimension:20 | | State Dimension:30 | |
| | | Failure times | Query times | Failure times | Query times | Failure times | Query times |
| $\epsilon = 0.1$ | BOGP | 0 | 3.04 ±0.83 | 78 | 849.26 ±295.35 | 3 | 86 ±200.49 |
| | GA | 0 | 64 ±0 | 0 | 65.92 ±10.92 | 8 | 183.04 ±287.80 |
| | PSO | 0 | 64 ±0 | 0 | 64.00 ±0 | 8 | 199.92 ±284.94 |
| | CMAES | 0 | 12 ±0 | 0 | 73.84 ±25.81 | 3 | 127.29 ±233.71 |
| | ISRES | 0 | 65 ±0 | 0 | 108.52 ±41.36 | 10 | 203.30 ±297.13 |
| | NSGA2 | 0 | 64 ±0 | 0 | 70.40 ±19.20 | 8 | 189.04 ±293.60 |
| | UNSGA3 | 0 | 64 ±0 | 0 | 68.48 ±16.33 | 7 | 177.52 ±275.84 |
| | SVPEN | 82 | 932.51 ±176.38 | 100 | 1000 ±0 | 100 | 1000 ±0 |
| | GEESE (ours) | 0 | **2.34 ±17.99** | 0 | **23.13 ±17.99** | 0 | **35.58 ±63.82** |
| $\epsilon = 0.05$ | BOGP | 0 | **9.24 ±3.97** | 100 | 1000 ±0 | 16 | 227.63 ±364.08 |
| | GA | 0 | 64.00 ±0 | 0 | 353.28 ±105.74 | 20 | 297.92 ±363.45 |
| | PSO | 0 | 64.00 ±0 | 1 | **157.84 ±137.40** | 18 | 290.96 ±373.65 |
| | CMAES | 0 | 77.56 ±4.38 | 1 | 302.59 ±156.24 | 22 | 344.54 ±363.18 |
| | ISRES | 0 | 193.00 ±0 | 3 | 391.54 ±241.22 | 19 | 313.69 ±368.78 |
| | NSGA2 | 0 | 64.00 ±0 | 0 | 352.00 ±114.31 | 20 | 299.84 364.63 |
| | UNSGA3 | 0 | 64.00 ±0 | 0 | 368.64 ±102.85 | 20 | 310.72 ±370.24 |
| | SVPEN | 100 | 1000 ±0 | 100 | 1000 ±0 | 100 | 1000 ±0 |
| | GEESE (Ours) | 0 | 20.20 ±16.37 | 0 | 189.90 ±164.96 | 2 | **81.26 ±155.30** |

case with lower error tolerance, and $\epsilon = 0.1$ representing a comparatively easier case with higher error tolerance. Results are reported in Table 4. In both cases, GEESE has failed the least times among all the compared methods and for all three problems studied. It is worth to mention that, in most cases, GEESE has achieved zero failure, and a very small $N_{\text{failure}} = 2$ out of 100 in only one experiment when all the other methods have failed more than fifteen times. Also, this one experiment is the most challenging, solving the most complex problem 3 with the highest state dimension $d = 30$ and having the lowest error tolerance $\epsilon = 0.05$. In terms of query times, GEESE has always ranked among the top 2 most efficient methods for all the tested cases and problems, while the ranking of the other methods vary quite a lot. For instance, when $\epsilon = 0.05$, BOGP performs the best for the easiest problem 1, but it performs the worst for the more difficult problem 2 where it has failed to find a feasible solution within the allowed query budget. In the most difficult experiment that studies problem 3 under $\epsilon = 0.05$, GEESE requires much less query times and is significantly more efficient than the second most efficient method.

**Varying Initial Sample Size:** In addition to the studied initial sample size $N = 64$ in the main text, we further compare to more cases of $N = 16$ and $N = 32$ under $\epsilon = 0.05$. The results are shown in Table 5. Still, GEESE has the least failure times in all experiments, which is important in remediating failed ML estimations. In terms of query times, GEESE still ranks among the top 2 most efficient methods for the two more complex problems 2 and 3, being the top 1 with significantly less query times for the most complex problem 3. However, GEESE does not show advantage in the simplest problem 1 with the lowest state dimension. It performs similarly to those top 2 methods under $N = 32$, e.g. 34 vs. 32 query times, while performs averagely when the initial sample size drops to $N = 16$. This is in a way not surprising, because BOGP, GA, PSO, NSGA2 and UNSGA3 can easily explore the error distribution of low state dimensions. BOGP uses Gaussian process to construct accurate distribution of errors, while GA, PSO, NSGA2, and UNSGA3 sample sufficient samples in each iteration to sense the distribution of error in each iteration, and there is a high chance for them to find a good candidate in early iterations when the search space has a low dimension. However, the valuable samples are sparsely distributed into the higher dimensional space, and it is challenging for them to explore the error distribution and find the feasible states in the early iterations.

## D    GEESE Sensitivity Analysis

We conduct extra experiments to assess the hyperparameter sensitivity of GEESE using problem 1 under $\epsilon = 0.05$. The studied hyperparameters include the number $L$ of the base neural networks, the number $N_{\text{IT}}$ of the candidate states generated for exploitation, the learning rate for training the exploitation generator $\eta_{\text{IT}}$, and the early stopping threshold $\epsilon_e$ for training the base neural networks. The results are reported in Table 6. It can be seen from the table that, although the performance varies versus different settings, the change is mild within an acceptable range. This makes it convenient to tune the hyperparameters for GEESE.

Table 5: Performance comparison under for two different sizes of initial samples, where the best is shown in **bold** while the second best is underlined for query times.

| Initial Size | Algorithm | Problem 1 State Dimension:11 | | Problem 2 State Dimension:20 | | Problem 3 State Dimension:30 | |
|---|---|---|---|---|---|---|---|
| | | Failure times | Query times | Failure times | Query times | Failure times | Query times |
| $N = 32$ | BOGP | 0 | **9.60 ±3.89** | 100 | 1000 ±0 | 15 | 239.12 ±367.12 |
| | GA | 0 | 32.00 ±0 | 0 | 241.60 ±71.75 | 21 | 270.80 ±382.51 |
| | PSO | 0 | 32.00 ±0 | 18 | 311.20 ±333.45 | 14 | 283.28 ±324.54 |
| | CMAES | 0 | 77.56 ±4.38 | 1 | 321.01 ±188.6 | 22 | 280.54 ±363.18 |
| | ISRES | 0 | 64.00 ±0 | 3 | 416.24 ±209.23 | 21 | 276.24 ±386.75 |
| | NSGA2 | 0 | 32.00 ±0 | 1 | 239.44 ±150.26 | 22 | 262.88 ±394.99 |
| | UNSGA3 | 0 | 32.00 ±0 | 2 | **218.72 ±136.53** | 22 | 260.64 ±396.51 |
| | SVPEN | 100 | 1000 ±0 | 100 | 1000 ±0 | 100 | 1000 ±0 |
| | GEESE (Ours) | 0 | 33.63 ±19.35 | 0 | 233.96 ±180.01 | 10 | **167.77 ±284.31** |
| $N = 16$ | BOGP | 0 | **10.62 ±5.53** | 100 | 1000 ±0 | 17 | 249.88 ±372.99 |
| | GA | 0 | 16.00 ±0 | 43 | 657.04 ±352.42 | 23 | 364.40 ±373.30 |
| | PSO | 0 | 32.00 ±0 | 10 | 293.76 ±271.02 | 21 | 247.76 ±392.87 |
| | CMAES | 0 | 77.56 ±4.38 | 1 | 333.49 ±156.24 | 17 | 320.07 ±350.84 |
| | ISRES | 0 | 17.00 ±0 | 2 | 260.50 ±189.71 | 20 | 243.20 ±392.70 |
| | NSGA2 | 0 | 32.00 ±0 | 33 | 590.96 ±355.93 | 25 | 377.20 ±385.78 |
| | UNSGA3 | 0 | 32.00 ±0 | 28 | 487.04 ±360.22 | 28 | 408.80 ±397.77 |
| | SVPEN | 100 | 1000 ±0 | 100 | 1000 ±0 | 100 | 1000 ±0 |
| | GEESE(Ours) | 0 | 36.72 ±22.52 | 0 | **248.26 ±176.64** | 9 | **163.26 ±279.34** |

Below we further discuss separately the effects of different hyperparameters and analyze the reasons behind: (1) We experiment with three base network numbers $L = 2, 4, 8$. It can be seen from Table 6 that there is a performance improvement as $L$ increases in terms of the required query times, but this is on the expense of consuming higher training cost. Thus, we choose the more balanced setting $L = 4$ as the default in our main experiments. (2) We test different candidate state numbers $N_{IT} = 1, 32, 64, 128$ used for exploitation. Results show a performance increase followed by a decrease as $N_{IT}$ increases. Using a candidate set containing one single state is insufficient, while allowing a set with too many candidate states can also harm the efficiency. This can be caused by the approximation gap between the surrogate error model and the true physical evaluation. In our main experiments, we go with the setting of 64 for problem 1 as we mentioned in Appendix B, because it provides a proper balance between the exploitation performance and the overfitting risk. (3) We also examine different settings of the learning rate for training the exploitation generator, i.e., $\eta_{IT} = 1e^{-1}, 1e^{-2}, 1e^{-3}$. Similarly, there is a performance increase first but followed by a decrease, as in changing $N_{IT}$. A larger learning rate can accelerate the learning of the exploitation generator and subsequently enable a potentially faster search of the feasible state. But an overly high learning rate can also cause fluctuation around the local optimum, and this then consumes more query times. Although a smaller learning rate can enable a more guaranteed convergence to the local optimum, it requires more iterations, thus more query times. (4) We experiment with three values of early stopping threshold, i.e., $\epsilon_e = 1e^{-3}, 1e^{-4}, 1e^{-5}$. It can be seen from Table 6 that a decreasing $\epsilon_e$ can first improve the efficiency but then reduce it, however without changing much the standard deviation. An inappropriate setting of the early stopping threshold can lead to base neural networks overfitting (or underfitting) to the actual data distribution, thus harm the performance.

Table 6: Results of sensitivity Analysis, where a better performance is highlighted in **bold**.

| (1): Effect of Base Network Number | |
|---|---|
| **Base Network Number** | **Query times** |
| $L = 2$ | 20.20 ±16.37 |
| $L = 4$ | 15.44 ±13.86 |
| $L = 8$ | **15.09 ±13.01** |

| (2): Effect of Latent Vector Number | |
|---|---|
| **Latent vector number** | **Query times** |
| $N_{IT} = 1$ | 72.56 ±36.13 |
| $N_{IT} = 32$ | 28.21 ±17.05 |
| $N_{IT} = 64$ | **20.20 ±16.37** |
| $N_{IT} = 128$ | 26.95 ±14.12 |

| (3): Effect of Learning rate for Exploration Generator | |
|---|---|
| **Learning Rate** | **Query times** |
| $\eta_{IT} = 1e^{-1}$ | 27.56 ±9.28 |
| $\eta_{IT} = 1e^{-2}$ | **20.20±16.37** |
| $\eta_{IT} = 1e^{-3}$ | 64.36 ±44.50 |

| (4): Effect of Early Stopping Threshold | |
|---|---|
| **Early stopping threshold** | **Query times** |
| $\epsilon_e = 1e^{-3}$ | 37.55±17.28 |
| $\epsilon_e = 1e^{-4}$ | **20.20 ±16.37** |
| $\epsilon_e = 1e^{-5}$ | 26.20 ±15.45 |

