# OpenReview forum: "Physics-Driven ML-Based Modelling for Correcting Inverse Estimation"
_NeurIPS.cc/2023/Conference — NeurIPS 2023 spotlight_

### Official Review · Reviewer_mU9t · 2023-07-05

**Soundness:** 3 good
**Presentation:** 2 fair
**Contribution:** 3 good
**Rating:** 5
**Confidence:** 3

**Summary:**

This paper builds on recent advances in black-box optimization for science and engineering inverse problems to find a satisfactory state with small physical error while decreasing the query times to the physical evaluation. The authors propose an error correction method GEESE, which is composed of a hybrid surrogate error model and a generative twin state selection approach. The hybrid surrogate error model is defined by both explicit and implicit errors, with the implicit error modeled as an ensemble of fully-connected neural networks. The generative twin state selection method adopts an exploration and exploitation approach, reducing the cost associated with physical error collection by not directly choosing states via a comprehensive space search. Experiments in real-world engineering inverse design settings demonstrate that GEESE decreases the number of queries needed to find satisfactory states with low error tolerance.

**Strengths:**

The paper introduces a novel correction algorithm, GEESE to find a qualified state within an error tolerance threshold after querying the physical evaluations as few times as possible. Overall architecture is novel, and a twin selection strategy, its key component that simultaneously propose a potentially near-optimal state and a potentially informative state, is effective to decrease the query times.
The experiments demonstrate that the proposed method outperforms the baselines in terms of both failure and query times.

**Weaknesses:**

It is still unclear how existed ML model in ML-branch (in Figure 1) is incorporated in GEESE algorithm. Even though a state estimated by the existing ML-model is used for activating GEESE algorithm, what else is the ML-model used for?  Is the model also used to generate initial data $D_{0}$? (If so, does the performance of the ML model affect the failure time and query time in the experiments?) If ML-branch is used only for activation of GEESE branch, what is the reason of having ML-branch? In this case, as long as there is a pair of observation and estimated state (not necessarily being a state estimated by ML-model), GEESE is available for the pair. In other words, GEESE is applicable to no matter what ML-models or estimators we have.

**Minor comments**

line 149: Taking advantage of the robustness of nsemble learning [52, 53] … (typo)



**Questions:**

Is there any theoretical evaluation between the estimated state that activates GEESE and a satisfactory state by GEESE, such as distance between them?

**Limitations:**

Yes, the limitations are discussed in the main text.

---

> ### Author Rebuttal · Authors · 2023-08-09
>
> We thank you for the insightful comments. We address the comments by grouping them into two categories: questions and discussion about weaknesses.
> ### Replies to questions
> We would like to sincerely thank the reviewer for this excellent question. Let us assume an observation vector $\mathbf{y}$, and during training, its associated state vector $\mathbf{x}$, and corresponding state vector estimate $\hat{\mathbf{x}}$. We would like to emphasize that during testing, when GEESE is deployed, the ground truth state vector $\mathbf{x}$ for an observation $\mathbf{y}$ is unknown, and thus, it is not possible to assess whether the estimated state $\hat{\mathbf{x}}$ is satisfactory or not by calculating a distance metric in state space, i.e., $d(\mathbf{x}, \hat{\mathbf{x}})$, where $d$ is a suitable distance metric.
>
> To remediate this, the error system including physical models and metrics is used to project the distance assessment in state space into the error space, i.e., $e(\hat{\mathbf{x}})=\phi(d(\hat{\mathbf{x}}, \mathbf{x}))$. Function $\phi$ is the known physical model/metrics. By doing this, although we acknowledge the existence of $\mathbf{x}$, it is masked in the error function. Notably, such a projection is nonlinear and depends on the physical process. Therefore, the distance in state space, although correlated to the error, is not directly proportional to the error. Theoretically, it is difficult to assess whether a state estimate is satisfactory or not via the direct measurement in state space. However, in practice, we can approximately assess it through an error tolerance threshold $\epsilon$, which is set empirically. Given a  sufficiently small $\epsilon$, when $e(\hat{\mathbf{x}})<\epsilon$, we can assume $d(\mathbf{x},\hat{\mathbf{x}})<k\epsilon$ for some constant $k$. Therefore, we can say that the state estimate $\hat{\mathbf{x}}$ is satisfactory if satisfying the condition of $e(\hat{\mathbf{x}})<\epsilon$. This thresholding approach is currently used to activate/continue the run of GEESE.
>
> There are certainly deeper answers to explore your very interesting question, which we intend to pursue as further work, specifically, to perform geometric learning in state space, studying the geometry (e.g., distance and manifold structure) in state space that is compatible to the physical error.
>
> ### Discussion about weaknesses
> **1** We agree with the reviewer in that GEESE can be applied within a general estimation framework, without the requirement of using a ML branch. As explained in the introduction part (Section 1), data-driven machine learning methods resulted in substantial advances in the solution of inverse problems (please check lines 25-27). However, these approaches lack in terms of reliability and trustworthiness, especially, for applications with low error tolerance. Indeed, this is the motivation for introducing GEESE as a state correction framework to enhance the reliability and trustworthiness of state estimations, while capitalizing on the efficiency of data-driven machine learning approaches. In regards to the query on the initial dataset $D_0$, this is randomly chosen as explained in line 3 of the algorithm pseudocode. Lastly, the reason for introducing the ML branch in Figure 1 was to emphasize that GEESE is a state correction framework for ML estimators. Following the reviewer's comment, we removed the branch in question, and revised Figure 1, as shown in the pdf attachment.
>
> **2** Thank you for pointing out the typo. We have corrected it in the revised manuscript.

---

> > ### Comment · Reviewer_mU9t · 2023-08-20
> >
> > Thank the authors for providing detailed explanation for my questions. The explanation on the difficulty in estimating distance is very easy to follow. However, it still lacks theoretical soundness, such as a reason why GEESE can reduce the number of correction times and physical evaluations as well as estimation on the distance between $\bf{x}$ and $\hat{\bf{x}}$. Also, now that I have read reviews and the authors have updated the figure 1 (thanks for the update anyway!), I’m a bit unsure if the term “correction” in the title still makes sense, because I don’t see which function in the GEESE algorithm is actually responsible for “correcting" $\bf{x}$ and the proposed method looks more like a method to solve inverse problems for physics estimation. It would be great if the authors could provide an insight on the use of “correction.”
> >
> > Regardless of the lack of theoretical soundness, the empirical results are significant and I’d be happy to support acceptance.

---

> > > ### Author Response · Authors · 2023-08-21
> > >
> > > We would like to thank the reviewer for supporting the acceptance of this work. In order to clarify the role of GEESE, we would like to highlight the context of its application. We commence our analysis by considering the use of an ML-based inverse model, which provides an estimate of the system state $\hat{\mathbf{x}}$ (at iteration 1). The estimated state is evaluated by the independent error module $E_p$, as shown in Eq. (1). When the disparity between $\mathbf{x}$ and $\hat{\mathbf{x}}$ exceeds an error tolerance threshold $\epsilon$, the state correction framework of GEESE comes into play, with the aim of correcting the state estimate, so that it is within the prescribed error tolerance (within a finite number of iterations). This process is explained in lines 228-232 of the original submission, however, it will be further emphasized in the revised draft. We acknowledge the reviewer's point of view that it is entirely possible for GEESE to be directly used as the inverse solver, however, this would impact on its convergence speed (i.e., starting from a random or zero state). Given the computational complexity of iterative optimization, and the deployment requirements of engineering applications (e.g., problem 3: Pulse-width Modulation of 13-level Inverters), an appropriate solution is one that capitalizes on the computational efficiency of ML-based inverse models, and the ad hoc utilization of GEESE for improving the reliability of system state estimation, as explained in lines of 27-31, 43-46. Thus, we feel that the use of the "correction" term in the title of our contribution is well-justified and conveys the context of using GEESE.
> > >
> > > We are grateful to the reviewer for appreciating the significance of our empirical results, which resulted from strategically selecting only two states for physical evaluations at each iteration, and the efficient training of the robust hybrid error model to support the selection process. These designs aim at a reduced number of physical evaluations. Development of a theoretical proof for the reduction of the number of queries times to the physical evaluator is planned as part of our future research, and will be highlighted in the revised draft.

---

### Official Review · Reviewer_aYMv · 2023-07-06

**Soundness:** 3 good
**Presentation:** 3 good
**Contribution:** 2 fair
**Rating:** 5
**Confidence:** 4

**Summary:**

This paper studies the problem of estimating states from observations (inverse problem). The problem is important since there are many applications such as engine design for aircraft. The current solution usually uses a black box simulator to simulate the observations given the estimated states from the model and compare them with the ground-truth observations to adjust the estimation of the state. This is based on the gradient-based method. However, the current state-of-the-art approach is time-consuming, in which it requires many quires and gradient update steps to find a good solution. As a result, this paper aims to increase the sample efficiency of such a method.

The proposed method is called "Generative Exploitation and Exploration guided by hybrid Surrogate Error (GEESE)". It consists of several parts: (1) a surrogate error model that consists of an ensemble of neural networks to provide a fast estimation of the physical error, and (2) a generative twin state selection algorithm that consists of two neutral networks for generating the state's distribution. The algorithm proceeds as follows: it first proposes an "exploitation" state that aims to search for a better solution that is near optimal, and then an "exploration" state that aims to explore other states that might lead to a better solution. The proposed method is tested in three real-world inverse problems and shows a competitive performance.

**Strengths:**

1. The writing is very clear, and the notation is great.

2.The method is simple and intuitive, and it shows good results based on Table 1.

**Weaknesses:**

1. The method requires the training data to train the neural networks to estimate the error. This would require a lot of computational cost and time to collect high-quality data.

2. The proposed method essentially tries to learn a "system dynamics model" using neural networks and exploits this neural network to do optimization. As a result, you do not need to query the simulator many times.

3. Following on the above point, this means that the quality estimation heavily relies on the performance of the hybrid surrogate model. How many datasets do you use? If I understand correctly, first you "pre-train" the model, and then use that for getting the results in Table 1?

4. There is no explanation about what are the tasks presented in Table 1. I knew it states that they are in the appendix, but the reviewers are not required to read the appendix. So it would be nice to include some detail about the tasks.

5. The observation dimension in the paper is quite low, compared to some prior works that estimate the state from image observations (https://proceedings.neurips.cc/paper_files/paper/2022/file/6d5e035724687454549b97d6c805dc84-Paper-Conference.pdf, https://arxiv.org/abs/2102.06794).

**Questions:**

1. In Table 1, there is a metric called query times. Could you please report the world-clock time in seconds? I am curious to see if these three problems simulator actually requires an hour/minute to run.

2. How will the model perform if the size of the observation is in the order of thousand?

**Limitations:**

See the weakness section. Overall, I think this paper needs some further discussion.

---

> ### Author Rebuttal · Authors · 2023-08-09
>
> We thank you for the insightful comments. We address the comments by grouping them into two categories: questions and discussion about weaknesses.
> ### Replies to questions
> **1**  Thank you for the question. In practice, different types of simulators exist for the same problem. For instance, for problem 1, a  simulator mentioned in [A] with high fidelity takes 10.3 seconds $\times$ 309 iterations  $\approx$ 53 minutes to obtain a converged simulation, while another simulator in [B] with even higher fidelity takes two weeks to get a converged simulation. Since we aim at a research scenario that attempts to search a feasible state without querying too much data and without setting a high standard on data quality, we choose to use faster simulators with lower but reasonable fidelity in our experiments. For the three studied problems,  each simulator takes no more than five seconds to run. Since the computing time varies when changing simulators, we report the performance by query times. We will include such information in the revised draft for better clarification.
>
> [A] Zhang, Xiaobo, Zhanxue Wang, and Li Zhou. "An integrated modeling approach for variable cycle engine performance analysis." Journal of the Franklin Institute 360.8 (2023): 5549-5563.
>
> [B] Claus, Russell W., and Scott Townsend. "A review of high fidelity, gas turbine engine simulations." Proc. 27th Int. Congress of the Aeronautical Sciences, Nice, France, 19-24 September 2010. Bonn, Germany: International Council of the Aeronautical Sciences, 2010.
>
> **2** Thank you for raising this question. An observation with thousands of dimensions will not affect the calculation much.  As we mentioned in lines 118-121, and demonstrated in the original Figure 1, an observation, denoted by $\mathbf{y}$, is used to calculate the error in GEESE and serves as a constant when correcting the state estimation. Its size only affects the error calculation, but not the optimization in state space. For instance, the choice of the distance function used to measure the difference between the ground truth observation $\mathbf{y}$ and the simulated one $\hat{\mathbf{y}}$ using the estimated state $\hat{\mathbf{x}}$ should depend on the observation nature and size.
>
> ### Discussion about weaknesses
>
> **1**  We agree with the reviewer that training data (state-error pairs in our study) is needed to train the surrogate error model (an ensemble neural network in our study). But our research objective is to reduce the cost of collecting high-quality training data, since we collect data online instead of having a ready dataset.  We collect a small amount of data to activate the training, and then incrementally collect as small amount of new data as possible at each iteration, aiming at effective online learning.  To avoid the use of a large readily collected data and to avoid high computational cost of data collection, GEESE uses a twin state selection approach to query the simulator at most twice in each iteration  (please refer to Sec.3.2). In experiments, for the studied problems, we only collect randomly 64 state-error pairs to activate the training (please refer to line 274), and by the time when a feasible state is found, the number of new state-error pairs incrementally collected is between 3-63  (see Table 1).
>
> **2**  Thank you for this comment. In GEESE, the surrogate model training (a learning process) and the optimal state search/exploration (a search process) are conducted in an alternating manner at every iteration. Such an incremental learning manner and an alternating nature do require to access the simulator many times. Specifically, unlike those conventional surrogate-model-based optimization mentioned in lines 85-98, the surrogate model in GEESE is not pre-trained using a readily collected large dataset, but trained from scratch  (please refer to sec.3.2), where, at each iteration, the states selected by the search process are used by the learning process to update the surrogate model. In addition, the surrogate model may provide unreliable estimations, which is critical for the applications of low error tolerance, as discussed in lines 95-98, so we also need to query the simulator to assess the quality of the corrected state estimation. Therefore, the simulator is necessarily queried multiple times.
>
> **3** We agree with the reviewer that the estimation quality heavily relies on the performance of the hybrid surrogate model. We would like to re-iterate that the surrogate model is not pre-trained via pre-collected datasets. Instead, the data is collected via online querying the simulator. As mentioned before, our training is performed incrementally and aims at using as few training data as possible, which, for the studied problems, include 64  randomly collected state-error pairs and another 3-63 carefully selected pairs.
>
> **4** Thank you for the useful suggestion. A task description will be incorporated after the first sentence of Section 4 in the revised draft.
>
> **5**  We are thankful to the reviewer for sharing these two papers. We would like to clarify that our research aims at reducing data usage through more effective online search strategies for optimization supported by a surrogate error model trained from scratch, and also aims at reducing data usage in surrogate model training by selecting more useful states to query. This is in contrast to the approaches reported in these two works, which are based on data-hungry, pre-trained models. As stated in our previous response under Question 2, the observation size is not of concern in the proposed approach. However, the state size matters, and this is discussed in lines 342-346, together with the potential limitations.

---

### Official Review · Reviewer_EBsP · 2023-07-07

**Soundness:** 3 good
**Presentation:** 3 good
**Contribution:** 3 good
**Rating:** 6
**Confidence:** 4

**Summary:**

This paper proposes a sample-efficient method for correcting failed states in surrogate inverse problems. This is achieved through error estimation of optimized surrogate states, where some computationally expensive errors are approximated with a neural network, while the simple errors are computed explicitly. Exploitation and exploration networks sample new states to verify errors for, finding feasible states that will not lead to failure. Several science and engineering applications are shown for the proposed method, and it outperforms the baselines for all tasks in query time and failure times.

**Strengths:**

Well written abstract, concise, showing core idea and significance in the field. The proposed method is well explained, and easy to understand. The authors did a thorough comparison with baselines, and ablation study to investigate the different components of the work.

**Weaknesses:**

1) Minor note: The introduction could be shorter, its purpose is very clear in that it leads to the contributions and methodology of the proposed work in the last paragraph, but there might be too much information leading up to that.

2) The D-dimensional state space first mentioned in Line 151 lacks a practical introduction, the reader might find it hard to imagine what such a state space could look like. Is it a meshed airfoil shape, is it the 3D position and velocity vectors of some object, etc. This is mainly important since in Line 183 the authors explain the generative model for a candidate set of these states, which is then again more difficult to understand without knowing what exactly these states are.

3) The implicit errors should be more thoroughly explained, it is unclear how "expensive to collect" is defined, and once again, a practical example of how these look like would be appreciated. Most important of all, it is unclear why these should be estimated with a surrogate in the first place. If it is due to computational complexity, an analysis should be done on how slow the computation is, and how much of a speedup the surrogate can offer, all the while showing the limitations of the surrogate in terms of generalizability, since the error estimation likely worsens when the explored states are far off from existing data.

**Questions:**

1) In line 99 for the related works paragraph on RL, is it correct to assume that there is no practical benefits of RL for inverse problems as opposed to the black box optimization mentioned in the paragraph above? At least there does not seem to be any advantages mentioned in this paragraph on RL for inverse problems. Is RL able to find better optima, but is just resource-intensive?

2) Why is the exploration network using a generative model at all? Is the state space poorly parametrized, i.e., too high dimensional? Otherwise the maximization in Eq. 10 could just be performed over all states, and not the generative model of the states.

3) Did the authors tune hyperparameters for the baselines as well for a fair comparison?

4) In Line 285 the authors simplified the setting of X=Z, is it correct to assume there is no generative model in this case?

**Limitations:**

Limitations are clear, although understanding why the surrogate for implicit errors is necessary should be elaborated. This was introduced in a too vague manner.

---

> ### Author Rebuttal · Authors · 2023-08-08
>
> We thank you for the insightful comments. We address the comments by grouping them into two categories: questions and discussion about weaknesses.
> ### Replies to questions
> **1** Thank you for starting this interesting discussion. From our perspective, the biggest difference between RL and black-box optimization (BBO) is that the emphasis of RL is to train a policy/agent model that could be powerful for all scenarios, while BBO tries to search for the optimal solution for a given scenario. Such a  general policy/agent model requires more iterations for training, which is more data and time consuming than BBO. In our experiments, we have compared SVPEN [43], an algorithm inheriting the idea of RL, with BBO (see Table 1). The results show that it does query more data. Although RL is not efficient as an online optimization tool, it can train powerful models. One can use GEESE to improve further an estimate state obtained by RL.  In addition, RL can tackle very complex inverse problems via the existence of the policy model, one example is alpha-tensor[A], whereas BBO may be not suitable. We will enrich the related work paragraph on RL in the revised draft.
>
> **2** Thank you for raising this question. We would like to point out that, in most practice, the state space is factually decided by the application, rather than being parameterized as an algorithmic choice. The reason to use a generative model is to enable a maximization over a sampled set of states rather than the whole state space,  as we aim at finding a largely disagreed state instead of the most disagreed state. As we empirically observed during algorithm development,  the inclusion of the most disagreed state can cause instability in search process, and being slower. The goal of performing exploration over the whole space is to find the most disagreed state, for which the exploration via disagreement (EVD) approach used by RL [57,58] serves as a tool by putting effort in training the exploration generator. We have conducted ablation study comparing EVD with our approach, which corresponds to the Ablation Study (2) in Table 2. The results show that our approach performs better with fewer query times.  We will explain the above motivation for using an exploration generator in the revised draft.
>
> **3** Yes, we did. Specifically, we tuned the kappa of utility function in Bayesian optimization, the learning rate of SVPEN, and the success rule of ISRES.  As for the hyperparameters in GA-style algorithms, such as population size and offspring size, they are fixed to the batch size of GEESE, because it is unfair for GA to use a much larger population size than GEESE, which will exhaust the query budget in one run. In addition, problem 1 is directly inherited from the paper of SVPEN [43], so we took the default setting of SVPEN for that. We will include the information in Section B of supplementary material.
>
> **4** Thank you very much for noticing this potential misunderstanding that could be caused by the expression of $X=Z$, and we apologize for not being clear in line 285. It means that we set the latent space $Z$ as the state space $X$ without using any neural network to transform in between. We randomly sampled an initial state sets $X_{IT}^{0}$ containing $K$ states sampled from the state space. The exploitation state is directly optimized iteratively, e.g., by a gradient descent approach, based on  Eq. (7) (its slightly modified expression is shown in our submitted pdf file). The different states in $X_{IT}^{0}$ are used to initialize the optimization and result in $K$ solution states. The one with the smallest objective function value is selected as the exploitation state.
>
> ### Discussion about weaknesses
>
> **1** We are happy to shorten the introduction in the revised draft.
>
> **2** When we generally introduce inverse problem and state in the beginning of the paper, we have given some practical examples of state in lines 21-23. But we agree that it will benefit the readers' understanding to use examples when explaining the method. As suggested, we will add the following text after the sentence in line 151: "*An example of such a state space is a $D=2$ dimensional space, where the two dimensions correspond to the temperature and concentration states from spectroscopy.  Another example is a state space of $D=11$ dimensions with each state corresponding to a design parameter for aeroengine design, for which we provide more details in Section 4 and Section A of supplementary material.*"
>
> **3** There are two types of implicit error that drive us to construct surrogate model to estimate them. As pointed by the reviewer, one type is the expensive error that takes time to calculate.  An example of this is a high-fidelity engine simulator in [B]   that takes two weeks to get a converged simulation. But, more importantly, a stronger drive for constructing surrogate models is the existence of those non-differentiable errors obtained from non-differentiable simulators, which mostly contain databases and maps. Examples of such simulators include those of spectroscopy [17] and of gas turbines [11].  The error gradient information is important for optimizing Eq.(6). Therefore, although there is a risk of obtaining inaccurate estimation, the surrogate model is used to estimate the errors and their gradient.  We do aim at improving the estimation accuracy in our design. For instance, we use ensemble learning to improve model robustness, and propose the twin state selection approach to query more useful states in order to improve the data quality for training the surrogate model. We will surely explain more on implicit errors and why estimating them in the revised draft, as suggested by the reviewer.
>
> [A] Fawzi, Alhussein, et al. "Discovering faster matrix multiplication algorithms with reinforcement learning." Nature, 2022.
>
> [B] Claus, Russell W., and Scott Townsend. "A review of high fidelity, gas turbine engine simulations.",  2010.

---

> > ### Comment · Reviewer_EBsP · 2023-08-10
> >
> > Thank you very much for the rebuttal. The responses do strengthen the paper, knowing what practical applications can be expected, the comparison between RL and BBO, and why implicit errors are very much needed. There are no more questions from my side, the score has been increased since the motivation for this paper is much stronger with this knowledge.

---

> > > ### Author Response · Authors · 2023-08-21
> > >
> > > Thank you very much for your positive feedback and for increasing the score.

---

### Official Review · Reviewer_Kiaj · 2023-07-12

**Soundness:** 3 good
**Presentation:** 3 good
**Contribution:** 3 good
**Rating:** 8
**Confidence:** 3

**Summary:**

The paper is very good and interesting - and on a relatively novel topic that has so far recently seen very less attention in the AI community, and is only starting to see more attention with the recent questions on fairness and trust in AI models. The proposed GEESE algorithm is interesting, the foundation of the paper is mathematically sound, and this reviewer appreciates the idea of the authors to use the algorithm to build trust and confidence in traditional AI models.

**Strengths:**

The contribution to the world of physics + AI combined is in itself very useful for the AI community. The major USP of the paper is its clear thought process on how in complex real-world engineering systems the error tolerance is often very low, and thereby, one can obviously not trust traditional optimisation techniques in ML. The idea of using physics-informed optimisation is appealing.

**Weaknesses:**

Only issue with the paper is its title - it is very generic. This reviewer would suggest the authors to make the title more specific (e.g. include the mention of ML models in the title or something similar to show that it pertains to utility in ML applications, as is the thought with NeurIPS!).

**Questions:**

N/A

**Limitations:**

The limitations have been clearly addressed in the conclusions section/discussions of the paper.

---

> ### Author Rebuttal · Authors · 2023-08-08
>
> Thank you very much for appreciating our work and the very useful suggestion. Indeed the title should link better with the machine learning community. The new title will be **"A physics-driven machine learning framework for correcting inverse estimation"**.

---

### Official Review · Reviewer_Rpgy · 2023-07-16

**Soundness:** 4 excellent
**Presentation:** 3 good
**Contribution:** 3 good
**Rating:** 8
**Confidence:** 4

**Summary:**

The article presents a new method of solving inverse problems using what they call is a grey box method. The introduce key concepts of using Generative methods to reduce the number of objective function invocations in an optimization problem. The presented results are very good and show a lot of promise for the techniques. Although there are a few points that are not addressed properly. They are mentioned in the sections below.

Another good thing with the method is that you can continue doing Exploration in Parallel, although this is not discussed in the paper. But on this point, I have other concerns that are mentioned in the Questions section.

**Strengths:**

The methodology is explained in detail. The appendices are used adequately to describe the required concepts and proofs in detail. The code is provided in the supplementary material, but I could execute it because not much guidance was provided on how to execute the code.

**Weaknesses:**

The method of explanation is not correct. In lines 126 to 130 authors have explained the algorithm briefly, which is not explained correctly. Important details are missing which creates misunderstanding. A better way would have been to introduce a complete sketch of the algorithm in the form of an algorithm that could be presented here. Then each part of the algorithm could be explained later, for example, Hybrid Neural Surrogate Error Models and Twin State Selection. Another way could be do first embed these fundamental concepts for the reader and then explain the algorithm in one go. Personally, I will recommend the first approach because that is iterative and reinforcing.

Following are some language-related issues.
- Line 83
   A grey-box setting is thus more suitable in practice, where ones do not
   [Correction]
   A grey-box setting is thus more suitable in practice, where one does not
- Line 141:
   followed by the twin state selection for charactering the probability
   [Correction]
   followed by the twin state selection for characterizing the probability
- Line 257
   for evaluatation are explained
   [Correction]
   for evaluation are explained
- Line 307
   general, hile outperforming other methods
   [Correction]
   general, while outperforming other methods


**Questions:**

Line 207, if you are not training the generator then there it can be sampled from an appropriate distribution.  How random weights are going to help you when they are not being trained?

**Limitations:**

The authors have mentioned the limitations of complex systems with a large number of states. They have mentioned that they will work on this problem in the future. They will look into how they can solve the problem of extended training time and data requirements.

---

> ### Author Rebuttal · Authors · 2023-08-08
>
> We thank you for the insightful comments. We address the comments by grouping them into two categories: questions and discussion about weaknesses.
> ### Replies to questions
> **Q1** Thank you  for  asking this interesting question. The purpose of the exploration generator  $\mathbf{G}_R$  is to randomly sample a set of candidate states to select one state from by applying the criterion in Eq. (10). This is an improved selection strategy  over the exploration via disagreement (EVD) approach commonly used by RL (see references [57,58] as in the submitted paper). The difference is that EVD selects from the whole space, while ours selects from the candidate set sampled by $\mathbf{G}_R$ to reduce computational cost. Theoretically, any probability distribution can be used as $\mathbf{G}_R$. To encourage diversity,  different distributions are used for   different iterations  by adopting a parametric probability distribution function and varying its parameters at each iteration. In our design, we  use neural network to formulate such a  parametric distribution and varying its parameters by using random network weights. An alternative option is to use the  multivariate Gaussian distribution  and assign different  mean vectors and covariance matrices at different iterations.  In our practice, the neural network based distribution is sufficient to encourage diversity and select an effective exploration state.
> We are happy to enrich the discussion of our exploration generator design in the revised manuscript.
>
>
> ### Discussion about weaknesses
>  **1** Thank you for the valuable suggestion. In lines 126 to 130, we started by explaining the commonly used iterative framework by classical black-box optimization techniques, instead of our proposed algorithm. It would be more clear to include a sketch of our algorithm after this.  We also prefer the first approach as suggested by the reviewer,  and will include an algorithm sketch in the revised manuscript and then expand on each algorithm part. We have prepared such a sketch and have attached it in the submitted pdf file as Algorithm 1.
>
> **2** Thank you for carefully reading our paper and helping improve the writing. We have made the corrections and polished the English in the revised manuscript.

---

> > ### Comment · Reviewer_Rpgy · 2023-08-17
> > **Acknowledgement**
> >
> > Thank you for clarification.

---

> > > ### Author Response · Authors · 2023-08-21
> > >
> > > Thank you very much for the acknowledgement.

---

### Author Rebuttal · Authors · 2023-08-09

Dear reviewers,

We are attaching a PDF file here for providing the information that we mentioned in separate rebuttals. Thank you very much.

---

### Decision · Program_Chairs · 2023-09-21

**Decision:**

Accept (spotlight)

**Comment:**

This paper presents a new method for solving inverse problems using a grey box method. A sample-efficient technique is used for correcting failed states in surrogate inverse problems. This is achieved through error estimation of optimized surrogate states, where some computationally expensive errors are approximated with a neural network, while the simple errors are computed explicitly. The proposed method addresses a problem that receives little attention within the AI community. The paper is well-written, and technically strong, and the proposed technique is novel. The experiments support the claims made in the paper.